EMBO
Molecular Medicine

# AAV-mediated expression of proneural factors stimulates neurogenesis from adult Müller glia in vivo

Marina Pavlou [ID][1], Marlene Probst [ID][1], Lew Kaplan[1], Elizaveta Filippova[2], Aric R Prieve[1], Fred Rieke[1] & Thomas A Reh [ID][1][✉]

## Abstract

**The lack of regeneration in the human central nervous system (CNS) has major health implications. To address this, we previously used transgenic mouse models to show that neurogenesis can be stimulated in the adult mammalian retina by driving regeneration programs that other species activate following injury. Expression of specific proneural factors in adult Müller glia causes them to re-enter the cell cycle and give rise to new neurons following retinal injury. To bring this strategy closer to clinical application, we now show that neurogenesis can also be stimulated when delivering these transcription factors to Müller glia using adeno-associated viral (AAV) vectors. AAV-mediated neurogenesis phenocopies the neurogenesis we observed from transgenic animals, with different proneural factor combinations giving rise to distinct neuronal subtypes in vivo. Vector-borne neurons are morphologically, transcriptomically and physiologically similar to bipolar and amacrine/ganglion-like neurons. These results represent a key step forward in developing a cellular reprogramming approach for regenerative medicine in the CNS.**

**Keywords** AAV Vectors; Müller Glia; Retina; Reprogramming; Neurogenesis
**Subject Categories** Genetics, Gene Therapy & Genetic Disease; Neuroscience

## Introduction

According to the World Health Organization, around 2.2 billion people worldwide suffer from vision impairment, with 415 million cases caused by diseases that affect the neuroretina, such as age-related macular degeneration, diabetic retinopathy and glaucoma (World report on vision). This has sparked global efforts to develop therapies that can treat retinal degeneration, with some success in the small molecule, gene supplementation, and gene editing fields for specific diseases caused by genetic mutations (Bainbridge et al, 2008; Song et al, 2022; Pierce et al, 2024; Russell et al, 2017; High and Roncarolo, 2019). In most cases, vision loss is caused by the irreversible death of key neuronal classes: cone photoreceptors and RGCs that are responsible for detecting light and relaying retinal output signals to the brain, respectively. Some organisms however, like fish and amphibians, are capable of regenerating their retina using resident cell populations that respond to injury (Reh and Nagy, 1987; Goldman, 2014; Parain et al, 2024). In fish, for example, the resident Müller glia (MG) upregulate key developmental factors in response to injury as they re-enter the cell cycle and dedifferentiate into multipotent progenitors that produce new neurons and restore retinal function (Wan and Goldman, 2016; Powell et al, 2016; Hammer et al, 2022).

Recently, there has been significant progress towards inducing neurogenesis in the adult mammalian retina by mimicking natural regeneration pathways from other species. It has been shown, using transgenic mouse models to drive the expression of developmental genes in adult MG, that these cells reprogram into proliferating neurogenic progenitors and neurons. Moreover, different combinations of transcription factors can stimulate MG to generate distinct neuronal lineages. For example, expression of achaete-scute family bHLH transcription factor 1 (Ascl1) alone stimulates neurogenesis of bipolar-like neurons, whereas combining Ascl1 with either atonal bHLH transcription factor 1 (Atoh1) or ISL LIM Homeobox 1 (Isl1) and POU domain class 4 transcription factor 2 (Pou4f2) stimulates genesis of neurons resembling bipolar, amacrine or retinal ganglion cells (Jorstad et al, 2017; Todd et al, 2021, 2022). Interestingly, the ability of mammalian MG to undergo proliferation and produce new neurons is not impacted by the timing or mode of injury, as the transgenic expression of Ascl1 or Ascl1-Atoh1 will stimulate neurogenesis after excitotoxic injury or light damage equally well, and the factors can be expressed before or after the injury with similar levels of neurogenesis (Pavlou et al, 2024).

Having identified combinations of factors that can push glial cells to generate different neuronal lineages, the next challenge is to establish a way to translate this biology into a medical application for regenerative medicine. This requires the delivery of exogenous proneural genes in mammalian MG and driving their

[1]Department of Neurobiology and Biophysics, University of Washington, Seattle, WA, USA. [2]Department of Agricultural and Biological Engineering, Purdue University, Lafayette, IN, USA. [✉]E-mail: tomreh@uw.edu

expression in a temporal manner. In this study, we investigate the feasibility of using viral vectors, specifically adeno-associated viral (AAV) vectors, to deliver the reprogramming factors to the MG and induce neurogenesis in vivo.

AAV vectors have become the gold standard for genetic perturbations in the eye, with several successful applications in the field of gene therapy against monogenic retinal disorders (Pierce et al, 2024; Russell et al, 2017; High and Roncarolo, 2019; Bainbridge et al, 2015; Li and Samulski, 2020). This has led to several previous efforts to test AAV-mediated strategies to stimulate neurogenesis with proneural genes driven by cell-specific promoters (Blackshaw and Sanes, 2021). Unfortunately, these early studies were confounded by the lack of glial specificity of the promoters that were used, particularly when driving the expression of proneural transcription factors. This led to the false conclusion that AAV-mediated reprogramming strategies led to neurogenesis, when in fact the promoter specificity was altered by the expression of transcription factors, leading to off-target AAV cargo expression in endogenous neurons rather than the induction of neurogenesis (Blackshaw and Sanes, 2021; Hoang et al, 2022; Le et al, 2022; Wang et al, 2021; Yao et al, 2018; Zhou et al, 2020). As such, we still lack convincing evidence that glia can proliferate and give rise to new neurons in response to AAV-borne expression of proneural factors.

To circumvent these artifacts, we used AAVs to deliver proneural factors in the adult mouse retina and a transgenic mouse line to restrict the AAV-payload expression to MG. Specifically, we used a transgenic mouse line (Rlbp1-CreERt2 x LSL-TdTomato) that enabled us to restrict Cre recombination only in MG in a tamoxifen-dependent manner, and thereby regulate the expression of the vector payload as well as permanently lineage trace MG and their progeny. In this way, the glial specificity is not conveyed by the AAV vector cassette but rather by the transgenic mouse line. This dual approach obviates the issues from prior studies, and allows us to test whether AAV-delivered proneural factors can stimulate neurogenesis from MG in the adult mouse retina in vivo.

To maximize our targeting efficiency of MG in the adult mouse retina, we delivered proneural genes using the AAV.7m8 vector, which was shown to be potent when delivered through the vitreous in vivo (Dalkara et al, 2013). Using a combination of histology, EdU labeling of newborn cells, scRNA-seq, and patch-clamp electrophysiology, we show for the first time that AAV-borne reprogramming factors can stimulate neurogenesis from MG after retinal injury. The efficiency of neurogenesis is affected by MG transduction, which is a function of both the vector titer and the incubation period of AAVs in the retina before inducing cassette expression. Interestingly, the types of neurons generated by the MG are dictated by the transcription factors expressed and the genetic trajectory of glia-to-neuron conversion phenocopies our results from previously reported transgenic mouse models of regeneration.

# Results

## Vector-borne Ascl1 expression stimulates MG proliferation and bipolar cell neurogenesis

In previous studies with transgenic mice, Ascl1 expression stimulated MG to re-enter the cell cycle and generate new

bipolar-like neurons after inner or outer retinal injury and the subsequent administration of HDAC inhibitor Trichostatin-A (TSA) (Jorstad et al, 2017; Pavlou et al, 2024). To determine whether AAV-borne expression of Ascl1 in adult MG phenocopies this response, we injected adult mice intravitreally with AAV.7m8 vectors expressing an inducible FLip-EXcision (FLEX) cassette, where a ubiquitous promoter (CBh or Ef1α) is upstream of the inverted Ascl1-TurboGFP cassette (Fig. 1A). As a control, we used an equivalent vector expressing only the fluorescent reporter.

To effectively lineage-trace MG and their progeny we used a transgenic line where Cre recombinase is expressed under the control of the Retinaldehyde Binding Protein 1 (Rlbp1) promoter and translocates to the nucleus only after exposure to tamoxifen, labeling MG with TdT (Rlbp1-CreERt2 x LSL-TdT) (Fig. EV1A). We have previously reported that this transgenic line restricts Cre and TdT expression to MG (Wohl et al, 2017; Wohl and Reh, 2016) and verified a mean ~70% recombination efficiency in Sox2$^+$ MG after 4-5 tamoxifen injections (Fig. EV1B) without any detectable colocalization with HuC/D$^+$ amacrine/retinal ganglion cells (Fig. EV1C–C") or Otx2$^+$ photoreceptors/ bipolar neurons (Fig. EV1D–D"). In the absence of tamoxifen, we observed minimal background signal of TdT in adult animals (2–3 months old), which was confined to Sox2$^+$ MG (Fig. EV1E–E', 43 TdT$^+$ cells on retinal flatmount).

To determine the optimal AAV incubation period, we injected adult Rlbp1-CreERt2 x LSL-TdT mice (that were not given tamoxifen) with AAV.7m8/Rlbp1-Cre and assessed TdT reporter expression at 2-, 4-, 6- and 8-weeks post injection (Fig. EV1F). Vector-treated eyes had detectable TdT signal in Sox2$^+$ MG already after 2-weeks incubation, which increased and plateaued after 4–6 weeks (Fig. EV1G–G'); we therefore concluded a 2-week incubation would be sufficient to obtain AAV-derived expression in the adult mouse retina. As such, after a 2-week incubation period of the Ascl1-expressing vectors in vivo, mice were given 5 daily doses of tamoxifen to trigger the recombination of the FLEX cassette and label MG with TdT (Fig. 1A). We evoked a damage response in the retina by injecting an excitotoxic dose of NMDA intravitreally, followed by an intravitreal dose of TSA 48 hours later (Fig. 1A). In our transgenic mouse studies, we allowed the animals to survive for 3 weeks after the injury and TSA; we therefore used a similar survival period for the AAV-derived Ascl1 expression. Three weeks after the NMDA/TSA injections, we sacrificed the animals and assessed the histology of the retina for evidence of AAV transduction (Fig. 1B), MG proliferation (Fig. 1C), and neurogenesis (Fig. 1D).

To ensure that our observations were not confounded by ectopic vector expression, we checked whether the FLEX cassettes had any traces of recombination during vector production or leakiness in the absence of tamoxifen. Both FLEX[GFP] and FLEX[Ascl1-GFP] vectors were injected intravitreally into Rlbp1-CreERt2 x LSL-TdT (n = 5 animals) and sacrificed 5 weeks later, consistent with the total duration of reprogramming experiments (Fig. EV2A). For the control FLEX[GFP] vector we detected rare GFP$^+$ cells that did not co-localize with neuronal markers Otx2 or HuC/D and had a MG morphology. For the FLEX[Ascl1-GFP] vector we did not detect clear GFP$^+$ cells. As noted above (see Fig. EV1E–E') there were rare Cre-independent TdT$^+$ cells in the red channel that had a MG morphology and did not co-localize with Otx2 or HuC/D (Fig. EV2A'). As such, we concluded that the FLEX cassettes used were reliable for our reprogramming studies.

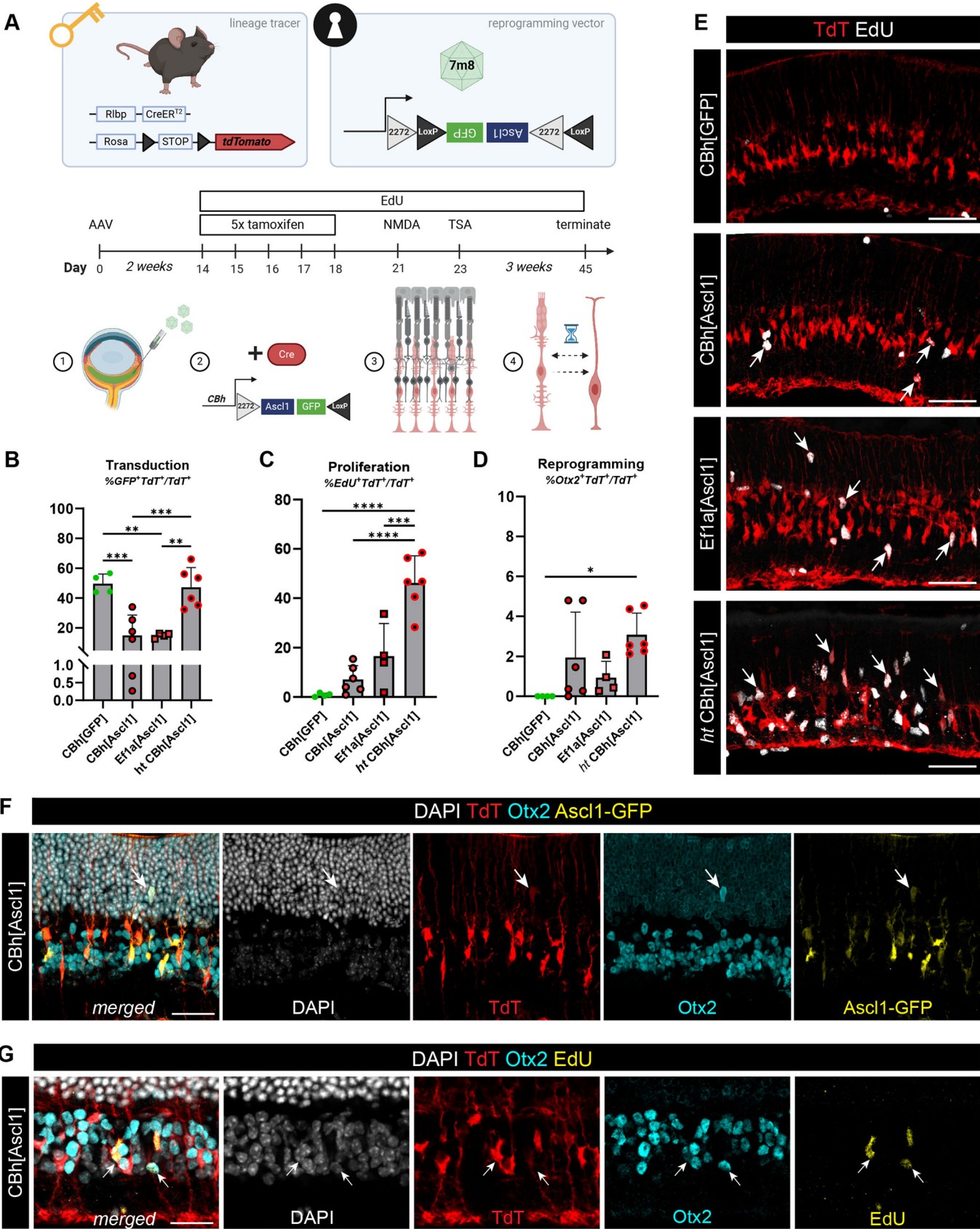

**Figure 1.  AAV-borne Ascl1 expression in MG stimulate their proliferation and neurogenesis of bipolar-like neurons.**

(A) Schematic of experimental concept and timeline of each intervention in vivo. (B) Bar plot of transduction efficiency per condition quantified as the percentage ratio of GFP+TdT+ cells over all TdT+ with each dot being a biological replicate. (C) Bar plot of MG proliferation counted as a ratio of EdU+TdT+ over all TdT+ cells with each dot being a biological replicate. (D) Bar plot of MG reprogramming counted as a ratio of Otx2+TdT+ over all TdT+ cells with each dot being a biological replicate. (E) Fluorescence images of MG proliferation per condition showing EdU in white and TdT in red, with double-labeled examples indicated with white arrows. (F, G) Fluorescence images of reprogrammed cells (white arrows) per condition showing merged and single channels of DAPI in white, TdT in red, Otx2 in cyan and Ascl1 or EdU in yellow; error bars for bar plots: mean plus standard deviation, statistical significance based on ordinary one-way ANOVA with Tukey's multiple comparisons test (significance $P < 0.05$, * = 0.05, ** = 0.01, *** = 0.001, **** = 0.0001), scale bar for (E): 50 μm, (F, G): 20 μm. Source data are available online for this figure.

In the initial round of experiments, we tested two strong ubiquitous promoters, Ef1α and CBh, with the latter at two different titers. Transduction efficiency was assessed by the number of GFP+TdT+/TdT+ cells in three regions across the retinas. The control vector transduced half of TdT+ MG (mean = 49%, $n = 4$ animals), the low-titer constructs with either CBh or Ef1α promoters driving Ascl1 yielded a modest transduction efficiency (mean = 10–15%, $n = 4$ animals), and the high-titer construct with the CBh promoter driving Ascl1 also transduced nearly half of TdT+ MG (mean = 47%, $n = 6$ animals) (Figs. 1B and EV3A).

We found that all three vectors driving Ascl1 triggered MG proliferation (Fig. 1C), with patches of TdT+ MG expressing EdU and with nuclei either in the inner nuclear layer (INL) or translocated across the apical-basal axis in the outer nuclear layer (ONL) or inner plexiform layer (IPL). These EdU+TdT+ MG were only found in the retinas treated with Ascl1, and none were present in the retinas of eyes injected with the control vector expressing only the fluorescent reporter (Figs. 1E and EV3A). The rare EdU+ cells detected in the control condition were not TdT+ and most likely microglia, as they localized in the plexiform or nerve fiber layers. As noted, the efficiency of transduction varied significantly across vectors (Fig. 1B), but irrespective of the range in transduction, all constructs expressing Ascl1 stimulated MG proliferation, with the high-titer vector having a significantly higher mean effect of 46% EdU+TdT+ MG ($n = 6$ animals) (Fig. 1C). We verified that this effect was indeed due to transduction and vector-derived expression of Ascl1, as GFP+TdT+ cells were also Ascl1+ and a subset of them was EdU+ (Fig. EV3B).

In addition to MG proliferation, vector-borne expression of Ascl1 stimulated the neurogenesis of Otx2+ bipolar cell (BC)-like neurons (Fig. 1D,F), as seen in our previous reprogramming studies with transgenic animals (Jorstad et al, 2017; Pavlou et al, 2024). The low-titer constructs driving Ascl1 yielded ~1–2% reprogramming, with the CBh promoter producing a slightly stronger but not significant effect compared to Ef1α (CBh mean = 1.9% versus Ef1α mean = 0.9%). This efficiency was nearly doubled with the high-titer construct where neurogenesis averaged 3% (Fig. 1D). Lineage-traced Otx2+ BC-like neurons were transduced (Fig. EV3C, white arrows), often mislocated in the ONL (Fig. 1F, white arrow), and a subset of them co-localized with EdU (Fig. 1G, white arrows), confirming they are bona fide newborn neurons.

To further characterize the effects of AAV-borne Ascl1 expression, we used fluorescence-activated cell sorting (FACS) to collect TdT+ cells pooled from several infected animals ($n \geq 3$) and then processed them for single-cell RNA sequencing (scRNA-seq) (Fig. 2A). Any residual cells that were not sequenced were stained confirming we sorted TdT+Sox2+ MG and TdT+GFP+Otx2+ BC-like cells from control and reprogrammed retinas, respectively

(Fig. EV4A–C). When plotting all the sequenced cells from the experimental conditions in a single integrated graph of reduced UMAP space, we could identify cell clusters based on common transcriptional profiles (Fig. 2B). Most of the cells were MG, some of which were reactive (Gfap+, Ifit3+, Stat1+), and smaller clusters were identified as microglia (Ptprc+, Tnf+, Aif1+), astrocytes (Pax2+, S100b+), rod (Nrl+), and cone photoreceptors (Arr3+) (Fig. 2B), most likely carried over during sorting. We have encountered this caveat of imperfect sorting in our previous experiments of transgenic reprogramming (Jorstad et al, 2017; Todd et al, 2022, 2021; Pavlou et al, 2024), which is why we verify bona fide neurogenesis via histology and EdU co-expression with neuronal markers. Since we do not observe EdU+ cells with definitive rod/cone markers, we cannot prove these were newly generated, and so do not include them in the further analyses of MG-derived neurons.

Of all the cells we analyzed, two cell clusters were distinct, in that they expressed MG transcripts as well as Ascl1 and its downstream targets Hes6, Myt1l and Otx2. These clusters were labeled Ascl1+ neurogenic precursors (NPre) and MG-derived BC-like neurons (BC-like) (Fig. 2B). Both clusters expressed genes AAVR (AU040320), Gpr108 and Tm9sf2 that are essential for AAV transduction (Summerford and Samulski, 2016; Pillay et al, 2016; Meisen et al, 2020) (Fig. 2B'). Cells in these clusters originated mostly from animals treated with Ef1α-FLEX[Ascl1] and high-titer CBh-FLEX[Ascl1] (Fig. 2C, orange and red), which confirmed our histological observations on proliferation and reprogramming (Fig. 1C,D). Interestingly, we observed no correlation between vector-borne neurogenesis and MG reactivity, as our control vector (CBh-FLEX[GFP]) accounted for the majority of reactive MG (Fig. 2C, dark gray). This is an encouraging indication that transducing MG with AAVs does not evoke an inflammatory response that prevents reprogramming, as we detect both proliferating MG (Fig. EV4E, TdT+EdU+ white arrows) and ramified microglia (Fig. EV4E, Iba1+EdU+ yellow arrows) in retinas transduced with CBh-FLEX[Ascl1].

The NPre uniquely expressed Ascl1 and the Notch ligand Delta1 (Dll1), whereas MG-derived neurons expressed the downstream effector Myt1l, neurogenic marker Neurod4 and bipolar marker Otx2 while retaining a low relative expression of the MG marker Rlbp1 in both clusters (Fig. EV4D). We also detected transcripts of neuronal markers expressed early during neurogenesis and often retained as neurons mature, such as Pcp4, Elavl3, Meis2, Caln1, Isl1, Ebf1, and Pou2f2 (Fig. EV4D).

The induction of neurogenesis in the MG we obtained with AAVs phenocopies our results from transgenic animals (Glast-CreERt2 x LSL-tTA x tetO-Ascl1-GFP), even though the glial promoter and mode of Ascl1 expression were different. Specifically,

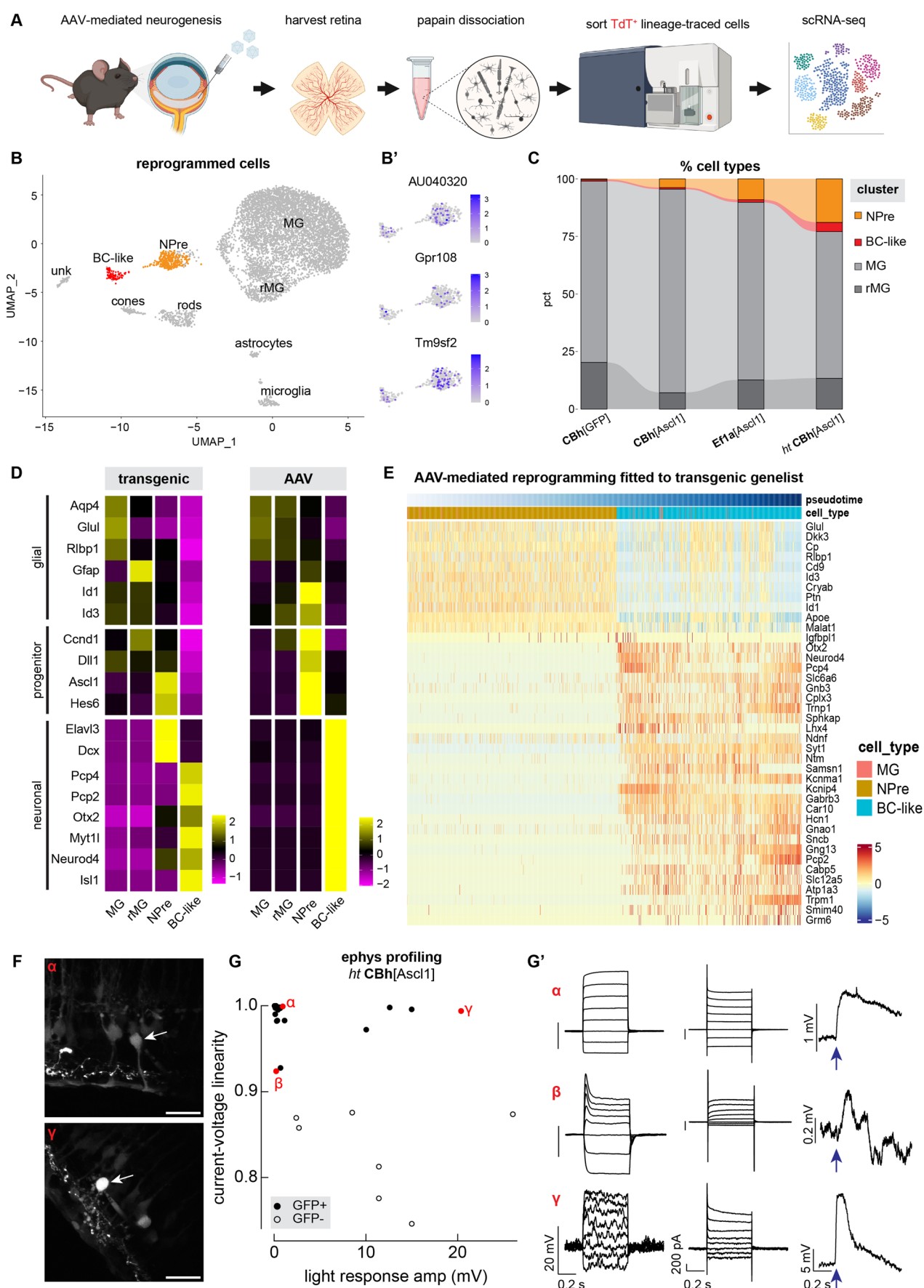

**Figure 2. Transcriptional and functional profiling of AAV-mediated reprogramming.**

(A) Schematic of the experimental pipeline. (B) UMAP of integrated scRNA-seq data from AAV-borne Ascl1 and control vectors clustered by cell type. (B') Feature plots of subset reprogrammed clusters showing the level of expression for AAV receptor genes. (C) Alluvium plot of each vector condition showing the proportion of clusters NPre, MG-derived bipolar cells (BC-like), MG and reactive MG (rMG) across samples. (D) Heatmaps of average gene expression level for select glial, progenitor and neuronal markers in rows and integrated scRNA-seq data clusters in columns from published transgenic (Jorstad et al, 2017; Pavlou et al, 2024) or AAV reprogramming experiments. (E) Heatmap of gene expression across pseudotime of AAV-mediated reprogramming fitted to the top 40 differentially expressed genes identified from transgenic scRNA-seq data plotted across pseudotime as they shift from glial to neurons. (F) Fluorescent images of representative cells (α and γ, white arrows) in live retina slices used for patch-clamp recordings. (G) Scatterplot of current–voltage linearity relative to light response amplitude for GFP$^+$ reprogrammed cells and GFP$^-$ control cells, from retina samples treated with the high-titer AAV/CBh[Ascl1] where red dots indicate example cells α β γ whose profiles are shown in (G'), (G') from left to right, family of voltage responses to current steps, family of current responses to voltage steps, and voltage responses to a brief light flash at the time of the arrow. Scale bar for (F): 20 μm. Source data are available online for this figure.

when we integrated the scRNA-seq data from either transgenic (Jorstad et al, 2017; Pavlou et al, 2024) or AAV-borne expression of Ascl1, we confirmed that the pattern of expression for known marker genes was similar between the two (Fig. 2D). The average expression of glial genes was highest in MG and reactive MG for both transgenic and AAV reprogramming, with STAT-pathway targets Id1/3 being upregulated in NPre of AAV-treated cells (Fig. 2D). This aligns with our previous observation that MG failing to generate neurons retain high levels of these inhibitor of differentiation genes (Jorstad et al, 2020) and may explain the low levels of Rlbp1 expression in the reprogrammed clusters.

Furthermore, we asked how similar the trajectory of reprogramming is between transgenic and AAV-mediated neurogenesis. To address this, we performed pseudotime analysis on an integrated scRNA-seq dataset from previous transgenic experiments (Fig. EV4F) (Jorstad et al, 2017; Todd et al, 2021; Pavlou et al, 2024) and our new AAV experiments. With pseudotime analysis, where the age/maturity of cells is ranked computationally based on their transcriptomic content, we chose the starting and ending nodes to track the glia-to-neuron conversion and queried which genes were differentially expressed across the trajectory. We obtained a gene list of top 40 genes that followed the trajectory of transgenic reprogramming (Fig. EV4G) and then asked how well the trajectory of AAV-mediated reprogramming fit that sequence of gene expression (Fig. 2E). Despite the differences in cell number that correspond to each cluster (MG, NPre, BC-like neurons) between transgenic and AAV experiments, the pattern of gene expression is mostly shared between the two modes of reprogramming. For example, we see consistent downregulation of glial markers Glul and Rlbp1 as cells transition to a neural progenitor state, as well as downregulation of the apolipoprotein gene ApoE and long noncoding RNA Malat1 (Figs. 2E and EV4G), which are primarily expressed by MG in the retina (Malek et al, 2005; Yao et al, 2016). Interestingly, the NPre during transgenic reprogramming upregulate Igfbpl1 (Fig. EV4G), which has been associated with axon regeneration (Guo et al, 2018); however, this is less prominent during AAV-mediated reprogramming (Fig. 2E). Overall, we found a 23% overlap between gene lists across pseudotime trajectories, confirming that vector-mediated reprogramming largely follows the transgenic trajectory when shifting between cell fates.

To assess whether newborn cells were functional and integrated in the existing retinal network, we performed patch-clamp recordings of single cells in live retina slices ($n = 3$ animals) from mice that were treated with the high-titer CBh-FLEX[Ascl1] vector (Fig. 2F, example cells that were dye-filled post recording: α and γ).

MG-derived GFP$^+$ cells are shown as filled circles, and open circles show the endogenous GFP$^-$ neurons. Responses varied considerably across recorded cells, with some cells resembling native MG while others approximating neuronal responses. For example, cell α in Fig. 2G has a near-linear current–voltage relation, characteristic of MG, as indicated by the similar spacing of the responses to current and voltage steps, and a small and slow light response. Despite these MG-like properties, these cells (in the upper left corner in Fig. 2G) had considerably higher input resistances than native MG ($280 \pm 80$ MOhm vs $40 \pm 17$ MOhm, mean ± SEM, 10 reprogrammed cells and 3 native MG), indicating a reduction in the K$^+$ currents that dominate the electrical properties of native MG. The current–voltage relations of some of the reprogrammed cells (e.g., cell β) were shaped by voltage-activated conductances, most likely K$^+$ currents. Current–voltage relations for all of the reprogrammed cells were more linear than those of endogenous neurons, indicating that expression of voltage-activate conductances in the reprogrammed cells did not reach normal neuronal levels. Of the 19 GFP$^+$TdT$^+$ cells we recorded, 8 had significant light responses, four that were quite small and four that were substantial. Interestingly, the largest light responses we obtained after AAV reprogramming (e.g., cell γ) are ~10× larger than previous transgenic experiments (Jorstad et al, 2017), and are similar to the largest light responses measured in endogenous neurons.

Consistent with their physiological responses, the reprogrammed clusters were enriched for both sodium and potassium channel genes. Ascl1$^+$ NPre expressed potassium channel genes Kcnq4 and Kcnh7 while the MG-derived neurons expressed sodium channel genes Scn1a, Scn2a, and potassium channel genes Kcna5, Kcnb2, Kcnd3, and Kcnh5 (Fig. EV4D). Furthermore, MG-derived neurons expressed genes for AMPA (Gria1, Gria2), metabotropic (Grm1, Grm2, Grm5, Grm6), and NMDA (Grin1, Grin2a) type glutamate receptors, as well as inhibitory ionotropic (Gabra3, Gabrb3) and metabotropic (Gabbr2) GABA receptors (Fig. EV4D). This suggests that AAV-borne reprogrammed neurons expressed machinery for both excitatory and inhibitory synaptic inputs, in line with their diverse electrophysiological properties.

## Vector-borne atonal factors synergize with Ascl1 to stimulate neurogenesis from adult Müller glia

We have previously shown that the combination of Ascl1 and the related proneural transcription factor Atoh1, was highly efficient in converting MG to a neurogenic state, with up to 80% of the

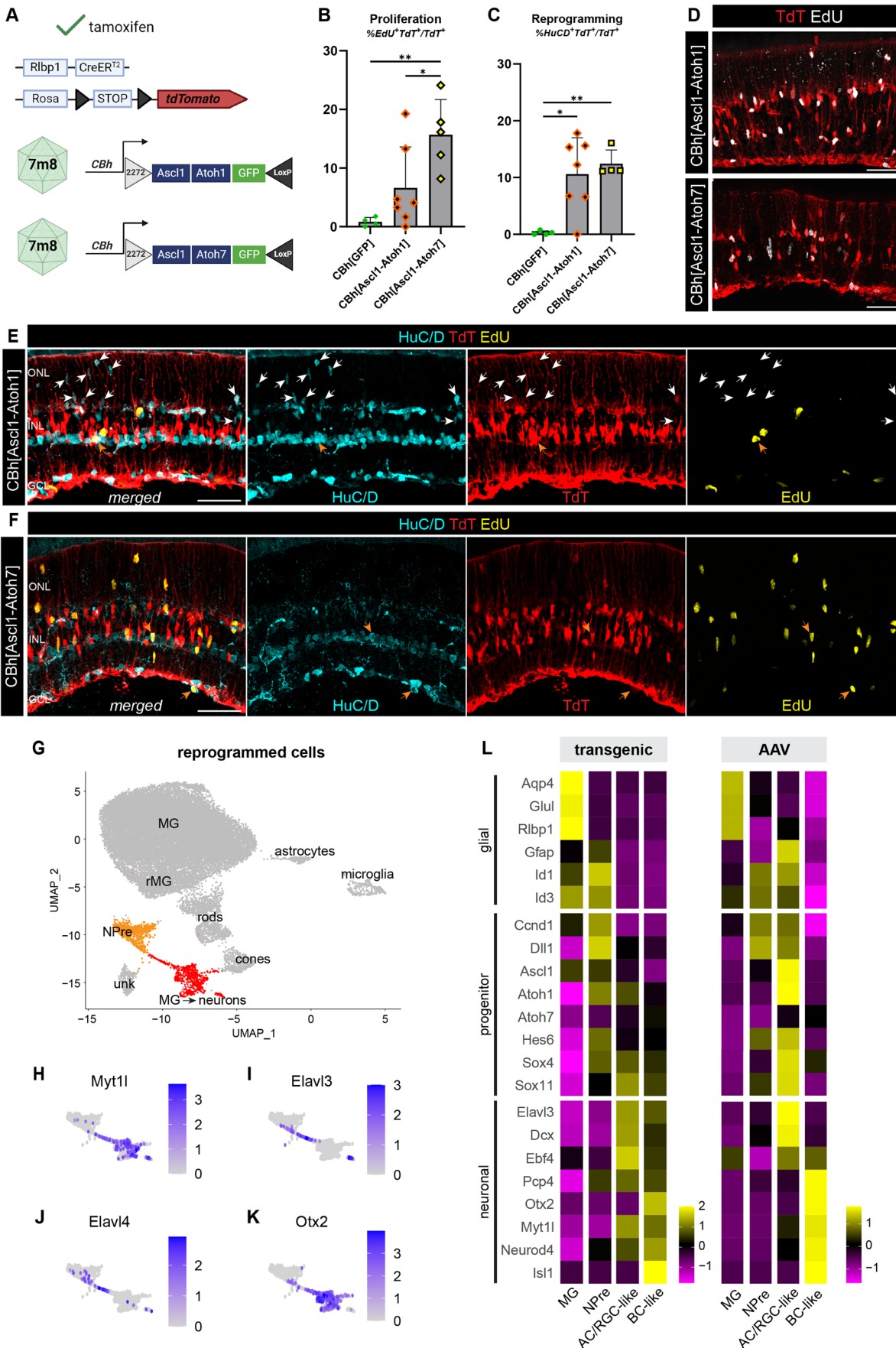

◀

**Figure 3.   AAV-borne Ascl1-Atoh1/7 expression induces neurogenesis that phenocopies transgenics.**

(A) Schematic of AAV vectors and their transgene cassette. (B) Bar plot of MG proliferation counted as a ratio of EdU$^+$TdT$^+$ over all TdT$^+$ cells with each dot being a biological replicate. (C) Bar plot of MG reprogramming counted as a ratio of HuC/D$^+$TdT$^+$ over all TdT$^+$ cells with each dot being a biological replicate. (D) Fluorescence images of MG proliferation per condition showing EdU in white and TdT in red. (E, F) Fluorescence images of reprogrammed cells (white and orange arrows) per condition showing merged and single channels of HuC/D in cyan, TdT in red, and EdU in yellow. (G) UMAP of integrated scRNA-seq data from AAV-borne Ascl1-Atoh1/7 clustered by cell type and highlighting reprogrammed clusters in orange/red. (H–K) Feature plots of subset reprogrammed clusters showing the level of expression for the gene annotated above each plot. (L) Heatmaps of average gene expression level for select glial, progenitor, and neuronal markers in rows and integrated scRNA-seq data clusters in columns from published transgenic (Pavlou et al, 2024; Todd et al, 2021) or AAV reprogramming experiments; error bars for bar plots: mean plus standard deviation, statistical significance based on ordinary one-way ANOVA with Tukey's multiple comparisons test (significance $P < 0.05$, * = 0.05, ** = 0.01, *** = 0.001, **** = 0.0001), scale bars for (D–F): 50 μm, ONL outer nuclear layer, INL inner nuclear layer, GCL ganglion cell layer. Source data are available online for this figure.

recombined MG generating neurons in transgenic animals irrespective of retinal injury (Todd et al, 2021; Pavlou et al, 2024). In addition, when Atoh1 was added to Ascl1 the MG-derived neurons adopted an amacrine cell (AC)/RGC-like fate rather than a BC-like fate. To determine whether similar increases in efficiency and changes in neural fate occur with vector-mediated expression of this transcription factor combination, we designed AAV vectors with a FLEX cassette of Ascl1 and Atoh1 or Atoh7 coding sequences driven off the CBh promoter (Fig. 3A). These two AAV vectors were intravitreally injected in adult Rlbp1-CreERt2 x LSL-TdT animals following the same experimental pipeline as described above (Fig. 1A).

AAV-mediated expression of Ascl1-Atoh1 in adult MG reproduced our observations from transgenic animals (Todd et al, 2021); we found an increase in MG proliferation and detected HuC/D$^+$TdT$^+$ MG-derived AC/RGC-like neurons. Since our vector payload was customizable and we were not limited by the availability of transgenic mice, we compared the combination Ascl1-Atoh1 ($n = 7$ animals) to Ascl1-Atoh7 ($n = 5$ animals) directly, which we previously reported to evoke neurogenesis from MG in vitro but had not yet tested in vivo (Todd et al, 2021, see Fig. 1).

Both combinations were able to induce robust MG proliferation (Fig. 3B) and neurogenesis in vivo (Fig. 3C), with ~5–15% lineage-traced MG incorporating EdU, similar to Ascl1 alone for the low-titer vector (Fig. 1C) and showing nuclear translocation across the basal-apical axis (Fig. 3D). For both vectors, driving Ascl1-Atoh1 or Ascl1-Atoh7, the main neuronal class obtained were HuC/D$^+$ AC/RGC-like neurons (Fig. 3E,F), with some Otx2$^+$ BC-like neurons generated as well (Fig. EV5A,B), in line with our previous report from transgenic reprogramming (Todd et al, 2021). Interestingly, Ascl1-Atoh7 was more efficient at driving the genesis of bipolar cells compared to Ascl1-Atoh1 (Fig. EV5B). Approximately 10% of lineage-traced MG gave rise to HuC/D$^+$ neurons after Ascl1-Atoh1/7 expression (Fig. 3C). Some of these cells were ectopically found in the ONL (Fig. 3E, white arrows), as we have seen in transgenic animals (Todd et al, 2021), and some incorporated EdU (Fig. 3E,F, orange arrows), thus confirming the cells were generated from proliferating MG.

Transcriptional profiling of TdT$^+$ sorted cells after AAV-mediated reprogramming confirmed our histological observations; we obtained distinct cell clusters of NPre and MG-derived neurons (MG→ neurons) (Fig. 3G). These clusters expressed Ascl1 but low levels of Atoh1/7 and TurboGFP, potentially due to the order of genes on the polycistronic vector cassette or transcript down-regulation. Importantly, we detected downstream effector markers such as Myt1l (Fig. 3H) and neuronal markers Elavl3/4 for AC/

RGC-like neurons (Fig. 3I,J) and Otx2 for BC-like neurons (Fig. 3K).

Vector-induced neurons phenocopy the fate and expression profile of glia-derived neurons from transgenic experiments (Todd et al, 2021; Pavlou et al, 2024), with known genes similarly patterned across MG, NPre, MG-derived AC/RGC-like neurons (AC/RGC-like) and BC-like neurons (BC-like) (Fig. 3L). AC/RGC-like neurons from transgenic animals seem to downregulate glial and progenitor genes more effectively compared to AAV, as the latter still express Gfap and Id1/3. However, for both conditions, AC/RGC-like neurons express Elavl3, Dcx, Ebf4, Sox4 and Sox11 (Fig. 3L), which are genes we previously identified in MG-derived AC/RGC-like neurons following transgenic expression of Ascl1-Atoh1 (Todd et al, 2021) and Islet1-Pou4f2-Ascl1 (Todd et al, 2022).

Our ability to stimulate neurogenesis using AAVs was most influenced by the vector titer and the promoter, though it was clear that the combinations Ascl1-Atoh1 and Ascl1-Atoh7 had a stronger effect than Ascl1 alone when delivered at the same titer (~2% for Ascl1, ~10–15% for Ascl1-Atoh1/7). This improvement in neurogenic potential was also true in previous experiments of transgenic models (Todd et al, 2021). Interestingly when comparing vector-mediated expression of Ascl1-Atoh1 and Ascl1-Atoh7, the latter seems more effective at inducing NPre and new neurons in our experimental context (Fig. EV5C), although the proportion of new neurons originating from proliferating EdU$^+$ cells was comparable across transcription factor combinations (Fig. EV5D). It remains unclear whether MG proliferation is required for neurogenesis, though in both the prior transgenic reports, as well as in the present examples of AAV-mediated reprogramming, we always find that some of the MG-derived neurons are not labeled with EdU, despite long periods of EdU administration.

We also performed pseudotime analysis to ask how similar the reprogramming trajectory is when Ascl1-Atoh1/7 are delivered via AAV as opposed to transgenics. To this end, we consolidated previous scRNA-seq datasets (Todd et al, 2021; Pavlou et al, 2024) and obtained a gene list of top 40 genes that followed the trajectory of transgenic MG reprogramming (Fig. EV5E). We then asked how well the trajectory of AAV-mediated neurogenesis fit that expression pattern. As we saw for Ascl1, the pattern of gene expression is mostly shared between the two modes of reprogramming when combining Ascl1-Atoh1/7 (Fig. EV5E). One obvious difference was the fluctuation of Atoh1 and Pcp4 in transgenic reprogramming, where they are first downregulated as the cells are in a glial state, then upregulated as they become progenitors and AC/RGC-like cells and again downregulated as they become BC-like. AAV-induced neurons do not follow the same pattern for

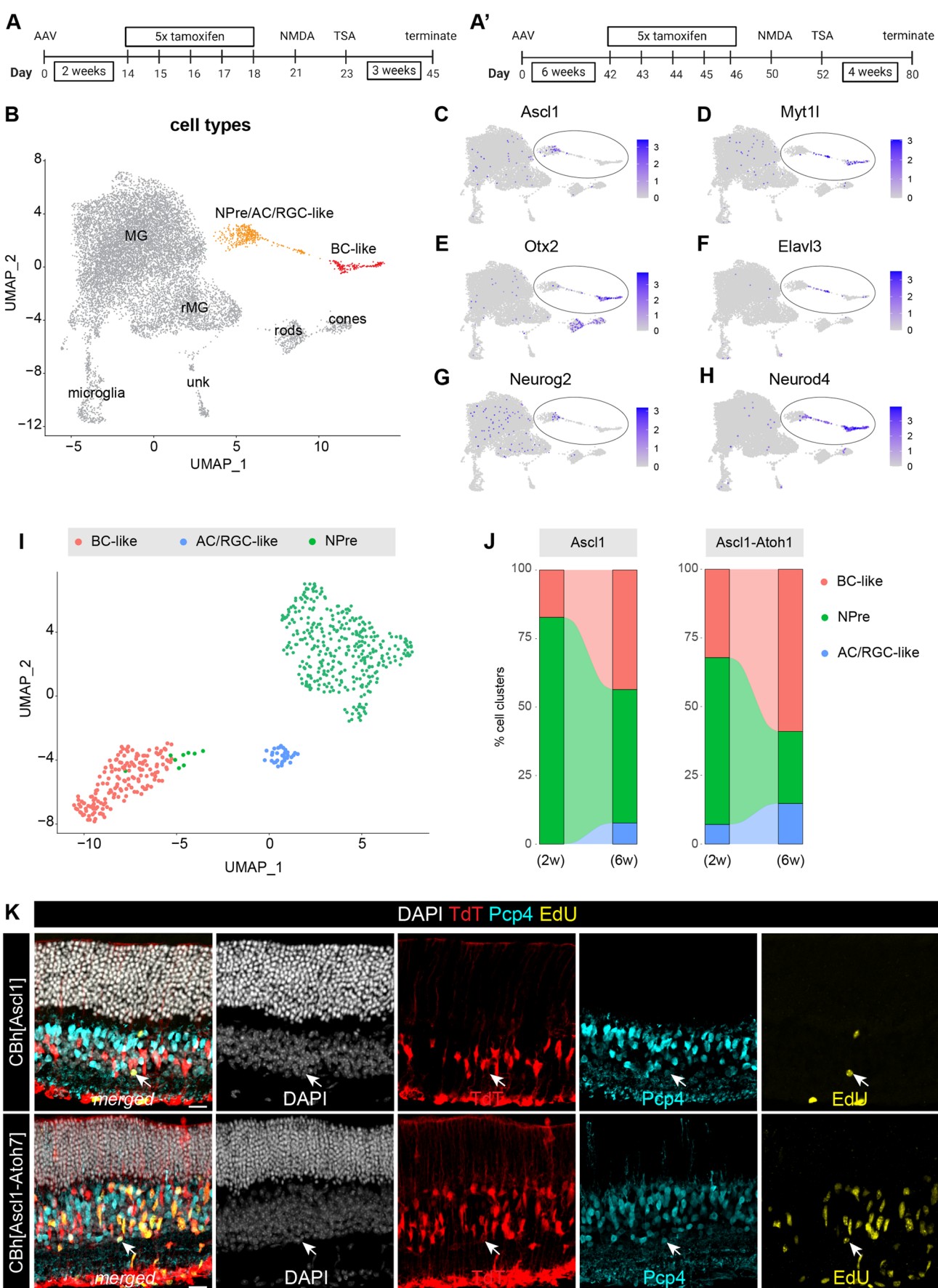

**Figure 4. The impact of AAV incubation time on neurogenesis.**

(A–A') Schematic of the experimental timeline where AAV is incubated for 2- or 6-weeks. (B) UMAP of integrated scRNA-seq data from AAV reprogramming experiments from all timepoints and vectors clustered by cell type. (C–H) Feature plots of integrated UMAP from (A) showing the level of expression for the gene annotated above each plot with reprogrammed clusters circled. (I) Reclustered UMAP of reprogrammed cells after integrating all AAV-borne reprogrammed cells for both timepoints. (J) Alluvium plots of each vector-derived cargo (Ascl1 or Ascl1-Atoh1) showing the percentage of each cell cluster in samples of 2- or 6-week AAV incubation from data subset shown in (I). (K) Fluorescence images of reprogrammed cells (white arrows) per condition showing merged and single channels of DAPI in white, TdT in red, Pcp4 in cyan, and EdU in yellow; scale bar: 20 µm. Source data are available online for this figure.

these genes, as Atoh1 levels seem consistent across states and Pcp4 is upregulated in the BC-like neurons. As such, vector-mediated reprogramming is not identical to transgenic animals, however overall, there is a similar trajectory of induced neurogenesis in vivo.

## Vector incubation affects the AAV-mediated reprogramming of distinct neuronal subtypes

The main rate-limiting step for AAV-genome expression is second-strand DNA synthesis to generate a double-stranded DNA molecule from which transcription can be initiated (Ferrari et al, 1996). As such, we hypothesized that a longer incubation period after AAV delivery in the retina would improve the expression of the vector cargo and by extension improve neurogenesis. To test this, we expanded our previous experimental pipeline and waited for 6 weeks after AAV injection (Fig. 4A–A') to match the previously reported transduction peak for capsid AAV.7m8 (Dalkara et al, 2013; Khabou et al, 2016). We used this expanded pipeline of AAV-borne Ascl1 or Ascl1-Atoh1 expression in transduced MG, followed by NMDA injury and TSA, and TdT$^+$ cells were subsequently sorted from retinal tissues 3-4 weeks post TSA (Fig. 4A').

We integrated the scRNA-seq datasets from all conditions, both for short (2-weeks) and long (6-weeks) incubation periods, and defined cell clusters based on their transcript profiles (Fig. 4B). The expression pattern of key genes Ascl1, Myt1l, Otx2, Elavl3, Neurog2, and Neurod4 (Fig. 4C–H) highlighted the reprogrammed cells (Fig. 4B, circled clusters). We then subset and reclustered these cells to specifically compare their distribution across timepoints, and profiled their three main states: cycling NPre, MG-derived BC-like cells, and MG-derived AC/RGC-like cells (Fig. 4I).

We found that the incubation period of the virus somewhat improved reprogramming efficiency. The 6-week incubation time samples, for both Ascl1 and Ascl1-Atoh1, showed an increase in the percentage of neurons (either AC/RGC-like or BC-like) and a relative reduction in the percentage of NPre cells when compared to the 2-week incubation time (Fig. 4J). This result is in line with the previously reported kinetics of the AAV.7m8 capsid (Dalkara et al, 2013) and our own AAV kinetics assay that reproduced the reported plateau of transduction after 6-weeks in vivo (Fig. EV1F,G). We did not observe any evidence that the neurons obtained following longer AAV incubation had a more mature profile, as we were able to obtain mature ON BC-like neurons (Pcp4$^+$TdT$^+$EdU$^+$) from both Ascl1 and Ascl1-Atoh7 vector-treated eyes after only 2-weeks of AAV incubation (Fig. 4K, white arrows). As such, a longer AAV incubation marginally improved reprogramming efficiency but did not affect the maturity of glia-derived neurons.

To better characterize how similar the reprogrammed cells are to endogenous neurons, we compared the transcriptome of MG-derived neurons from all AAV-infected timepoints with three published retinal datasets (Clark et al, 2019; Hoang et al, 2020; Li et al, 2024). Note, our sorting of TdT+ cells before scRNA-seq excluded endogenous bipolar, amacrine and retinal ganglion cells in our experiments. We compared bipolar subtypes from the Hoang et al dataset from adult mice post NMDA injury and find that the MG-derived BC-like cells upregulate genes that correspond to all BC subtypes in the mouse retina (Fig. 5A). We also compared the gene expression of the AC/RGC-like cells with that of mature retinal neurons and find that these are less comparable than the BC-like cells (Fig. 5B). However, when we make the same comparisons with P14 retinal neurons, we find neurons derived from the reprogrammed MG overlap well with the bipolar, amacrine and RGC clusters, but not MG (Fig. 5C,D). Interestingly, in both the Ascl1 and Ascl1-Atoh1/7 reprogrammed MG, there is a cluster of cells that do not align with any of the retinal neurons at P14. When reprogrammed cells are instead aligned with the E14 retina, they align primarily with progenitors, transitioning cells (T0 and T1), and the amacrine/photoreceptor branches (Fig. 5E,F). This solidifies our conclusion that vector-mediated expression of proneural genes in adult mouse MG can stimulate neurogenesis of distinct neuron classes, though not all these MG-derived neurons fully differentiate within the period of time we have analyzed.

## Discussion

The ability to stimulate neurogenesis from MG with transient expression of transcription factors provides a potential path to restoring vision in people that have lost retinal neurons due to injury or disease. Up to now, we have relied on transgenic expression of proneural factors, individually or in combinations, to study this approach in vivo. These mouse models were instrumental to our understanding of what neuronal classes we can obtain after expressing Ascl1 alone (BC-like neurons) or in combination with Atoh1, Islet1 and Pou4f2 (BC, AC and RGC-like neurons) (Todd et al, 2021; Jorstad et al, 2017; Todd et al, 2022). We were able to decipher important prerequisites for translation, using transgenic animals, such as the importance of retinal injury type and timing relative to proneural factor expression (Pavlou et al, 2024). To our advantage, MG expressing Ascl1 or Ascl1-Atoh1 can give rise to new neurons following both inner and outer retinal damage, after the neurons have already died (Pavlou et al, 2024); a necessary condition for treating most retinal diseases.

To generate a regenerative strategy that can someday be used to treat human retinal damage, one approach is to introduce

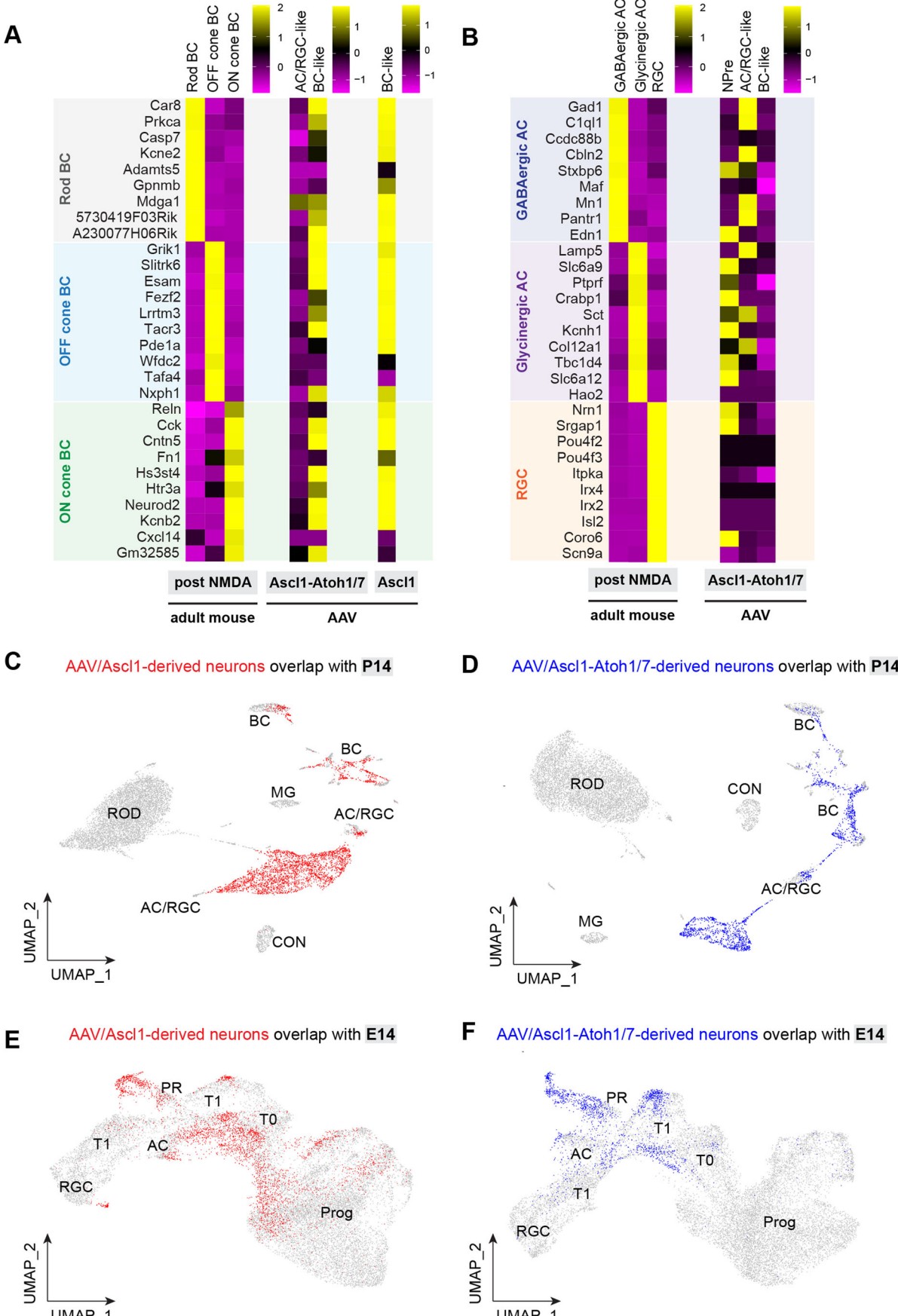

**Figure 5.  Comparative analysis of AAV-borne neurons and endogenous neurons.**

(A) Heatmap of top ten differentially expressed genes for BC subtypes in adult mouse retina post NMDA (Hoang et al, 2020) and their corresponding expression pattern across AAV-reprogrammed cell clusters grouped by vector treatment. (B) Heatmap of top 10 differentially expressed genes for AC/RGC subtypes in adult mouse retina post NMDA (Hoang et al, 2020) and their corresponding expression pattern across AAV-reprogrammed cell clusters grouped by vector treatment. (C, D) Integrated UMAP of scRNA-seq data from P14 mouse retina (Li et al, 2024) and AAV-reprogrammed cells showing the distribution overlap for (C) AAV/Ascl1 and (D) AAV/Ascl1-Atoh1/7 vector treatment across early postnatal cell clusters. (E, F) Integrated UMAP of scRNA-seq data from E14 mouse retina (Clark et al, 2019) and AAV-reprogrammed cells showing the distribution overlap for (E) AAV/Ascl1 and (F) AAV/Ascl1-Atoh1/7 vector treatment across developing cell clusters.

reprogramming factors to MG via AAV vectors. AAV vectors are small icosahedral capsids (~25 nm) that encapsulate a single-stranded (~4.7 kb) or double-stranded (~2.4 kb) DNA cassette flanked by inverted terminal repeats (ITRs) that encode the information for vector packaging and facilitate intermolecular recombination to form circularized concatemers for episomal persistence of AAV genomes in the host cell nucleus (Yang et al, 1999; Wang et al, 2019). AAV vectors are currently used in several clinical trials in the eye for gene replacement or gene editing therapies, and can drive long-term expression of AAV payload after just a single administration (Bainbridge et al, 2015; Li and Samulski, 2020).

Although AAV-mediated gene delivery has gained in popularity for ocular indications, there are several questions that need to be answered with regard to their use for MG reprogramming as a strategy for retinal regeneration: (1) can AAV-derived expression of proneural genes induce MG proliferation and neurogenesis, (2) can we restrict AAV vector expression to MG specifically, (3) what are the kinetics of AAV expression and does it matter, (4) will AAV vectors trigger a gliotic response that blocks reprogramming? In this report, we have tested these features and found that AAV-mediated reprogramming of MG is both traceable and effective.

Our results show that driving proneural gene expression in MG, where tamoxifen-inducible Cre expression is under the control of a transgenic Rlbp1 promoter, can result in high levels of recombination in the adult mouse retina and is sufficient for AAV-mediated expression of FLEX cassettes driven by ubiquitous promoters (CBh and Ef1α). This results in a robust effect of MG proliferation only when Ascl1, Ascl1-Atoh1, or Ascl1-Atoh7 are expressed, but not when a fluorescent reporter is expressed. Importantly, vector-mediated expression of these proneural genes led to neurogenesis towards a BC and AC/RGC-like fate following a similar gene trajectory as we previously reported in experiments with transgenic animals, even though the glial promoters in each case were different, namely Rlbp1 and Glast. The neurons we obtained after AAV-mediated reprogramming were integrated in the retinal network, had electrophysiological responses to light and expressed genes of the machinery required for both excitatory and inhibitory synaptic inputs. Strikingly, the maximum light response amplitude we obtained from several of these cells were much larger than reprogrammed cells from our previous transgenic experiments (Todd et al, 2022; Jorstad et al, 2017), and approached amplitudes characteristics of endogenous neurons. Thus AAV-mediated reprogramming produced cells that effectively integrated into their retinal circuit. Electrically, however, these cells differed less from MG than those we characterized in transgenic experiments; specifically, voltage-activated conductances in cells from AAV-

mediated reprogramming were considerably smaller than those from transgenic experiments.

With regards to specificity, recent reports have claimed AAV-mediated neurogenesis when in fact there was ectopic expression of vector payload in endogenous neurons or insufficient controls of vector specificity (Chen et al, 2020; Yao et al, 2018; Zhou et al, 2020). This has been addressed by follow-up studies that demonstrated the artifacts and highlighted caveats of using AAVs to induce cell fate changes, both in the brain and retina (Hoang et al, 2022; Wang et al, 2021). One of the causes for ectopic AAV-cargo expression was the use of the GFAP mini promoter, used to drive expression in astrocytes and MG (Chen et al, 2020; Yao et al, 2018; Zhou et al, 2020). The specificity of the GFAP promoter was carefully examined and shown to leak depending on the genes downstream, especially when driving proneural transcription factors such as NeuroD1, Ascl1 and Atoh7 (Le et al, 2022). In the same report however, FLEX cassettes under the control of a ubiquitous Ef1α promoter that drove proneural genes to MG did not result in leaky expression of AAV cargo in endogenous neurons, but also showed no evidence of induced neurogenesis from MG (Le et al, 2022). In our study, we show that restricting AAV-payload expression in MG can be achieved when the specificity originates from a transgenic Rlbp1 promoter. Importantly, this specificity does not change over longer periods of AAV incubation (2–8 weeks), even when the Rlbp1 promoter is delivered via AAV. In fact, adapting the experimental conditions to the optimal AAV kinetics (from 2-weeks to 6-weeks incubation) can slightly improve reprogramming efficiency.

We have shown that delivering Ascl1 or Ascl1-Atoh1/7 to MG with AAVs and controlling their expression with the FLEX system, can stimulate neurogenesis in MG in vivo and produce BC-like and AC/RGC-like neurons analogous to previous transgenic experiments. As with our previous studies, a subset of MG-derived neurons originated from proliferating EdU$^+$ MG, though it remains unclear whether proliferation is required for neurogenesis. Even if MG can directly transdifferentiate into neurons, we would argue that MG proliferation is beneficial for maintaining the MG population intact even after regeneration. Only MG that expressed proneural genes re-entered the cell cycle, and any inflammation evoked by the AAV injection, transgene expression, as well as the intravitreal NMDA and TSA injections, did not prevent neurogenesis with our paradigm. We show that the reactivity of MG was comparable across vector treatments, even at a high AAV titer, and similar in gene profile to the equivalent cluster of reactive MG from transgenic experiments. In large animal studies on non-human primates and human clinical trials, immunosuppression is administered prior to intraocular AAV injections as standard practice (Willett and Bennett, 2013; High and Roncarolo, 2019), which we envision would also become part of a future AAV-based regenerative regimen.

This body of work serves as a bridge towards our ultimate goal of an AAV strategy to stimulate retinal regeneration by expressing proneural genes in MG of adult patients. We dissected the issues of traceability and efficiency into a transgenic and vector component respectively, to demonstrate that AAV-borne expression of proneural factors is capable of altering cell fate in the adult mammalian retina in vivo. Now the challenge is to improve the efficiency of neurogenesis and design a strategy that is solely based on AAV-derived genetic material. By incorporating lessons learned from previous studies and the ongoing research on AAV biology, we can devise a reliable strategy that stimulates neurogenesis from MG using one or multiple viral vectors (Pavlou and Reh, 2023). Such a regimen would have an immense therapeutic impact and could be applied in a disease-agnostic manner. We believe this work makes a significant contribution to achieving this goal as it proves we are able to generate new neurons in an adult mammalian retina using vector-delivered cargo.

# Methods

### Reagents and tools table

| Reagent/resource | Source | Identifier | Concentration |
|---|---|---|---|
| Vector cassette | | Capsid | Titer GC/ml |
| CBh>FLEX[EGFP]-WPRE | Vectorbuilder | 7m8 | $2.15 \times 10^{12}$ |
| CBh>FLEX[hASCL1(ns):T2A:TurboGFP]-oPRE | Vectorbuilder | 7m8 | $3.93 \times 10^{11}$ |
| CBh>FLEX[hASCL1(ns):T2A:TurboGFP]-oPRE | Vectorbuilder | 7m8 | $4.60 \times 10^{13}$ |
| Ef1α > FLEX[hASCL1(ns):T2A:TurboGFP]-oPRE | Vectorbuilder | 7m8 | $8.44 \times 10^{11}$ |
| CBh>FLEX[hASCL1(ns):hATOH7(ns):P2A:TurboGFP]-oPRE | Vectorbuilder | 7m8 | $7.92 \times 10^{11}$ |
| CBh>FLEX[hASCL1(ns):T2A:hATOH1(ns):P2A:TurboGFP]-oPRE | Vectorbuilder | 7m8 | $5.62 \times 10^{11}$ |
| Experimental models | Reference | Identifier | |
| RLBP1-CreERt2 x LSL-TdTomato (M. musculus) | Wohl and Reh (2016) | N/A | |
| Antibodies | Source | Identifier | Dilution |
| Chicken anti-GFP | Abcam | AB13970 | 1:1000 |
| Mouse anti-HuC/D | Invitrogen | A-21271 | 1:200 |
| Goat anti-Otx2 | R&D Systems | BAF1979 | 1:500 |
| Rabbit anti-Ascl1 | Abcam | AB211327 | 1:300 |
| Rabbit anti-Pcp4 | Invitrogen | PA5-52209 | 1:1000 |
| Goat anti-Sox2 | Santa Cruz | SC-17320 | 1:200 |
| Donkey anti-chicken 488 | Jackson Immuno | 703-545-155 | 1:500 |
| Donkey anti-goat 405 | Invitrogen | A48259 | 1:500 |
| Donkey anti-goat 568 | Invitrogen | A11057 | 1:500 |
| Donkey anti-goat 647 | Jackson Immuno | 705-605-147 | 1:500 |
| Donkey anti-mouse 488 | Jackson Immuno | 715-605-150 | 1:500 |
| Donkey anti-mouse 568 | Invitrogen | A10037 | 1:500 |

| Reagent/resource | Source | Identifier | Concentration |
|---|---|---|---|
| Donkey anti-mouse 647 | Jackson Immuno | 715-605-150 | 1:500 |
| Donkey anti-rabbit 488 | Invitrogen | A21206 | 1:500 |
| Donkey anti-rabbit 568 | Invitrogen | A100042 | 1:500 |
| Donkey anti-rabbit 647 | Invitrogen | A31573 | 1:500 |
| Software | | | |
| ImageJ | https://imagej.nih.gov/ij/index.html | | |
| GraphPad Prism 10 | https://www.graphad.com | | |
| R Studio | https://cran.r-project.org/ | | |

## Animals

All animals were treated and housed with University of Washington Institutional Animal Care and Use Committee approved protocols. Males and females were both used in experiments at equal frequencies. All experiments were performed on adult mice that were over 30-days-old. For EdU incorporation, animals were given 0.4 mg/ml EdU/H2O *ad libitum* from the first day of intraperitoneal tamoxifen injections onwards until the animals were terminated.

## Injections

Intravitreal injections were performed with a 32-G Hamilton syringe on mice anesthetized with isoflurane. AAV preps were produced by Vectorbuilder. Intravitreal AAV injections were done in a volume of 1.5–2 µl and the vector titers are outlined in the reagents table. Intravitreal injections of NMDA were done in a volume of 1 µl at a concentration of 100 mM in PBS. TSA (Sigma-Aldrich) was administered via intravitreal injections in DMSO at a concentration of 1 µg/µl. Intraperitoneal injections of tamoxifen (1.5 mg per 100 µl of corn oil) were administered for five consecutive days to induce recombination of the AAV.7m8/FLEX cassette and transgenic LSL-TdTomato.

## Immunohistochemistry

Dissected eye cups were fixed with 4% paraformaldehyde/PBS for 30 min and then incubated in 30% sucrose solution at 4 °C overnight. Eyes were then embedded in optimal temperature cutting compound before freezing. Frozen samples were sectioned at −20 °C in 15- to 18-µM sections onto glass slides. Slides were then heated for 10 min on a slide warmer before staining or freezing at −20 °C for long-term storage. For staining, slides were traced with a liquid blocker pen and then rehydrated with PBS. For EdU detection the Click-iT™ EdU Imaging Kit (Thermo Fisher Cat # C10086) was used following manufacturer instructions, followed by antibody staining. Primary antibodies were incubated overnight at 4 °C in blocking solution (0.5% TritonX-100 and 5% Normal Horse Serum in PBS). The primary solution was removed, and the slides were washed with PBS. Secondary antibodies were incubated in blocking solution for 90 min and then slides were washed with PBS. Fluoromount-G (SouthernBiotech) mounting medium was added to slides before covering slide with a glass coverslip. See the table for all antibodies used.

## Microscopy

Stained sections were imaged using a Zeiss LSM800 microscope. For quantification, images were taken with a 20x objective with at least 3 images taken per retina section. Images were analyzed and counted using FIJI. Lineage-traced TdT$^+$ cells were initially counted from maximum intensity z-projections. The colocalization of markers was then identified and verified by going through single-planes of the z-stack. Careful attention was paid to ensure there was a consistent pattern of marker overlap in each plane where the cell was present. Individual performing cell counting was blinded whenever possible.

## Fluorescence-activated cell sorting

Following euthanasia, pools of four retinas were dissociated using the Worthington Papain Dissociation System (catalog no. LK003150) according to the manufacturer's instructions. Cells were then spun at 4 °C at $400 \times g$ for 10 min and resuspended in neurobasal medium (Gibco no. 21103049), 10% fetal bovine serum (Clontech), B27 (Invitrogen), N2 (Invitrogen), 1 mM L-glutamine (Invitrogen), and 1% penicillin-streptomycin (Invitrogen). The cell suspension was passed through a 35-µm filter and then sorted using a BD FACSAria III cell sorter (BD Bioscience) to retrieve TdT$^+$ cells. The FACSAria III cell sorter used was from the DLMP Flow Cytometry Core at the University of Washington. During sorting, appropriate gates were implemented to exclude debris and doublets. A minimum of 40,000 events were collected in tubes that were coated with 10% BSA.

## Electrophysiology

Recordings were performed identical to our previous reports (Todd et al, 2022). Mice were dark-adapted before recordings. After euthanasia, retinas were sliced into 200-µm slices for recording. Tissue recordings were performed in Ames medium at 32 °C and oxygenated with 95% $O_2$/5% $CO_2$. GFP$^+$ cells were targeted for recording using video differential interference contrast with infrared light and two-photon confocal microscopy. Light responses were measured in response to full-field illumination via a green light-emitting diode focused on the slice through the microscope objective. Recordings were performed using pulled glass pipettes and filled with solution containing the following: 123 mM K-aspartate, 10 mM HEPES, 1 mM MgCl2, 10 mM KCl, 1 mM $CaCl_2$, 2 mM EGTA, 0.5 mM tris–guanosine triphosphate, 4 mM MG–adenosine triphosphate, and 0.1 mM Alexa Fluor 488 hydrazide. Current–voltage relations were measured by imposing a series of current steps and measuring the resulting changes in voltage; linearity of these current–voltage relations was measured by the $R^2$ value of a straight line fit to the data.

## scRNA library construction

FACS-purified TdT$^+$ MG cells were spun at 4 °C at $400 \times g$ for 10 min and resuspended at a concentration of approximately 1000 cells/µL. Library construction was performed using the Chromium Next GEM Single Cell 3′ version 3.1 (dual index) protocol and reagents according to the manufacturer's instructions.

### The paper explained

**Problem**

Retinal neurons die as a result of aging and disease, such as age-related macular degeneration, glaucoma and diabetic retinopathy, leading to permanent loss of vision. This poses a major socioeconomic issue with growing patient numbers worldwide. So far, there are medical treatments to halt or decelerate the loss of neurons, but there is currently no therapy available that can regenerate neurons in the adult retina.

**Results**

We delivered proneural genes in the resident glia (Müller glia) of the adult mammalian retina, using adeno-associated viral vectors, and after injuring the retina we were able to show that these glia proliferated and gave rise to new neurons. These new neurons were verified by stringent lineage tracing, histology, single-cell sequencing and electrophysiology, to ensure accurate interpretation of neurogenesis and not artefactual detection of endogenous neurons, as seen in previous studies.

**Impact**

This is the first report to confidently show that vector-mediated expression of proneural genes in retinal Müller glia is sufficient and able to drive neurogenesis in the adult mammalian retina. Different combinations of proneural genes were able to stimulate the genesis of bipolar, amacrine, and retinal ganglion cell-like neurons, which could have therapeutic implications for diseases where these retinal subtypes are lost. This work serves as a premise for further optimization of vector strategies that could become a gene therapy for regenerative medicine.

## scRNA-seq, mapping, and data analysis

Multiplexed libraries were sequenced using an Illumina NextSeq 500 using high-output 150 kits. Data were demultiplexed and aligned to the mouse mm10 genome using Cell Ranger Count version 7.2 (Zheng et al, 2017). Filtered output files were further analyzed in R using Seurat version 4.3.2 (Hao et al, 2021), ggplot2, data.table, dplyr, tidyr, and other commonly used R packages. Low-quality cells (identified as having low read depth or high mitochondrial content; >10%) were removed from datasets. Before analysis, the cell number was down-sampled to the object with the lowest cell number, and cell unique molecular identifiers were downsampled to the lowest median for each object using the package scuttle's downsampleMatrix function. Cells were clustered using principal-components analysis and UMAP. Comparisons between datasets were made by canonical correlation analysis, as described by the Satija laboratory vignette (https://satijalab.org/seurat). Alluvial bar plots were created by the percentage of cells in each cluster split by each sample using ggplot2 with ggalluvial. Heatmaps was calculated using Seurat's DoHeatmap over the top differentially expressed (DE) genes or selected genes for each cell type cluster with their average expression of that gene. The top DE genes were found by using FindAllMarkers and sorted by the average log_twofold change for each single cluster compared against all others. Pseudotime-based trajectory analysis was conducted on the integrated RDS files using Monocle3 according to the Trapnell laboratory vignette (https://cole-trapnell-lab.github.io/monocle3/). The Seurat objects were converted to Monocle3 objects using 'as.cell_data_set()'. Cells were reclustered using 'cluster_cells()', and the principal graphs were calculated using 'learn_graph()'. Cells were ordered in pseudo-time using 'order_cells()' with the start nodes chosen based on

expressions of marker features. Paths on the graphs were selected using 'choose_graph_segments()' by specifying the start and end nodes. The paths were subset using Seurat's 'subset()'. The cells were reclustered, and the graphs were relearned in Monocle3. Gene lists of the top 40 differentially expressed (DE) genes were identified based on the highest Moran's I values obtained from 'graph_test()'. Expression of the top DE genes was plotted using ComplexHeatmap's 'Heatmap()'.

## Data availability

Some of the scRNA-seq analysis was done using previously published data and are referenced accordingly. All sequencing data are deposited to an online database GEO accession number GSE285980.

The source data of this paper are collected in the following database record: biostudies:S-SCDT-10_1038-S44321-025-00209-3.

## Peer review information

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

## Acknowledgements

The authors would like to thank all the members of the Reh lab and the Bermingham-McDonogh lab for their valuable comments on the manuscript. We would also like to thank Edward Levine for the transgenic mouse line RLBP1-CreERt2. All schematics were generated using Biorender. This work was funded by a postdoctoral fellowship from Weill Neurohub to MP, a Gilbert Family Foundation's Vision Restoration Initiative LLC grant to TAR, a National Institutes of Health grant NEI R01EY021482 to TAR and a sponsored research agreement from Tenpoint Therapeutics Ltd to TAR.

## Author contributions

**Marina Pavlou**: Conceptualization; Investigation; Visualization; Methodology; Writing—original draft; Writing—review and editing. **Marlene Probst**: Investigation; Visualization; Methodology. **Lew Kaplan**: Investigation. **Elizaveta Filippova**: Investigation; Visualization. **Aric R Prieve**: Investigation. **Fred Rieke**: Investigation; Visualization. **Thomas A Reh**: Conceptualization; Resources; Supervision; Writing—original draft; Writing—review and editing.

Source data underlying figure panels in this paper may have individual authorship assigned. Where available, figure panel/source data authorship is listed in the following database record: biostudies:S-SCDT-10_1038-S44321-025-00209-3.

## Disclosure and competing interests statement

Some of the findings in this report are part of a patent application that has been submitted by the University of Washington: Patent Application 63/362,361 filed 4 January 2022. TAR is a co-founder of Tenpoint Therapeutics Ltd. The remaining authors declare no competing interests.

# Expanded View Figures

**Figure EV1.  Assessment of transgenic lineage tracer mouse line and AAV kinetics in vivo.**

(A) Schematic of tamoxifen-inducible transgenic mouse line for MG-specific expression, (B) bar plot of min to max quantification of recombination efficiency counted as percentage ratio of RFP$^+$Sox2$^+$ cells over all Sox2$^+$ where each dot is a biological replicate (error bar: mean plus standard deviation), (C–C″) fluorescence images of Rlbp1-CreERt2 x LSL-TdT retina cross-sections after tamoxifen showing DAPI-stained nuclei in white, TdT in red and HuC/D in cyan, (D–D″) fluorescence images of Rlbp1-CreERt2 x LSL-TdT retina cross-sections after tamoxifen showing DAPI-stained nuclei in white, TdT in red, Otx2 in blue and EdU in yellow, (E) fluorescence image of an untreated retina flatmount with TdT in red and vasculature in white, (E′) representative retina cross-section of untreated retina with TdT in red and Sox2 in yellow (F) schematic of experimental design, (G) fluorescence images of retina flatmounts for each timepoint with TdT in red and vasculature in white, (G′) representative retina cross-sections from timepoints in (G) with DAPI-stained nuclei in white, TdT in red and Sox2 in yellow; scalebars for (C–C″, D–D″, G′, E′): 50 μm, for (E, G): 500 μm, ONL outer nuclear layer, INL inner nuclear layer, GCL ganglion cell layer.

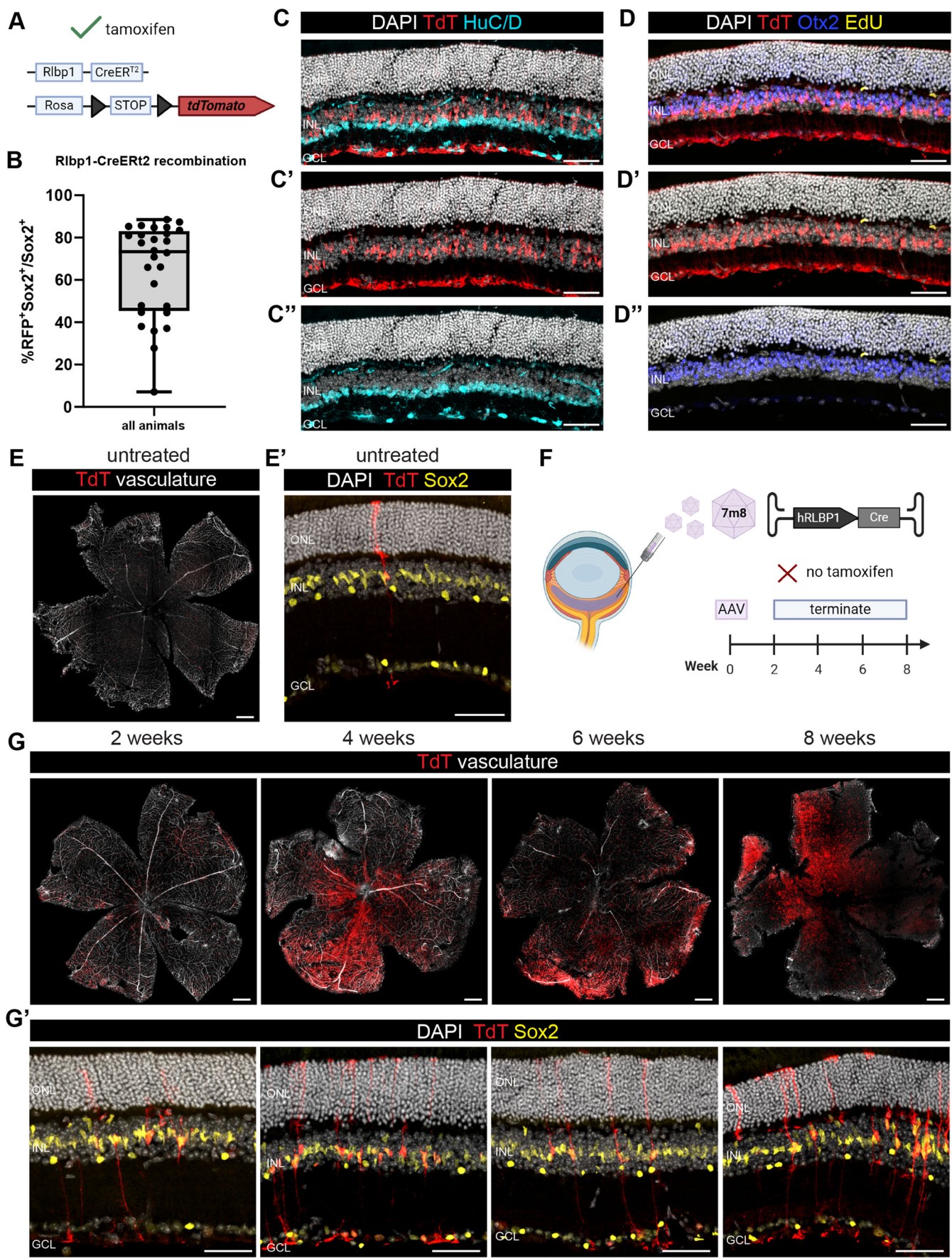

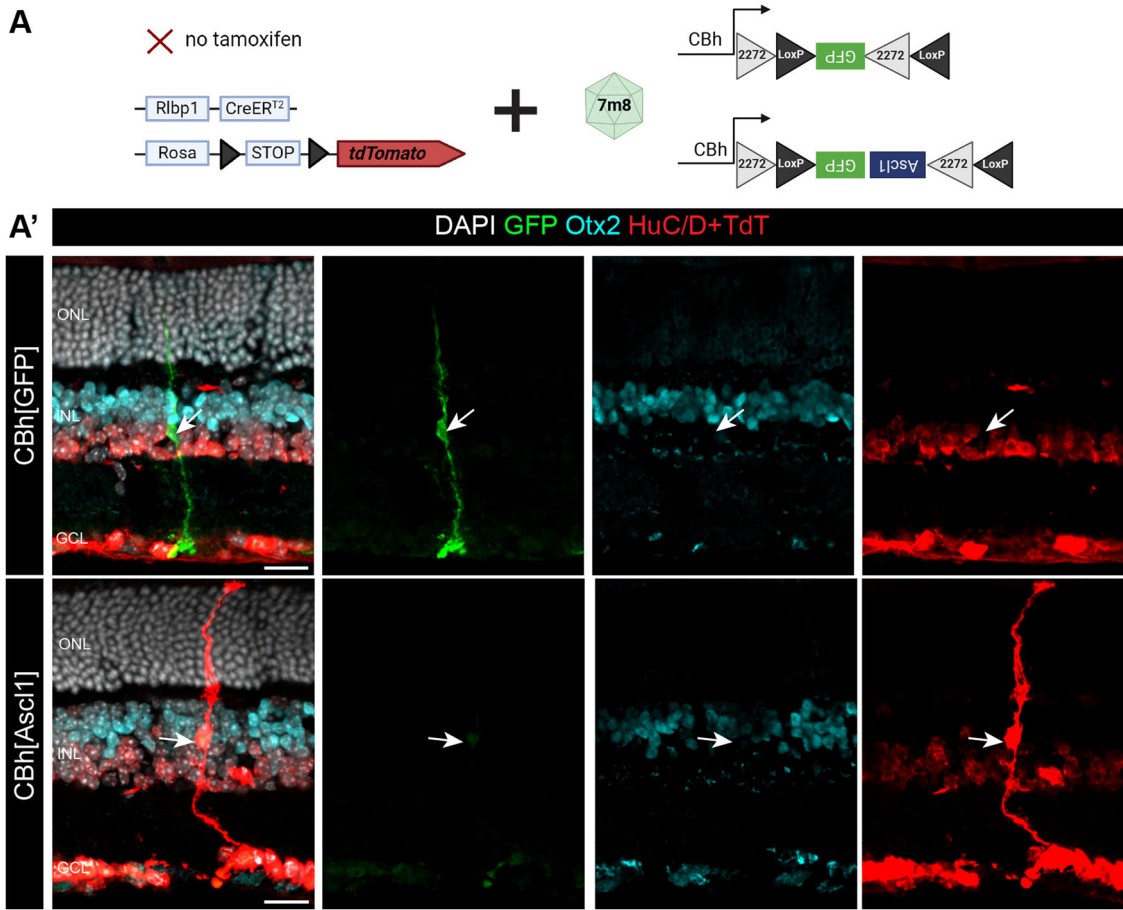

**Figure EV2.  Control experiments to assess Cre-dependent vector expression.**

(**A**) Schematic of transgenic lineage tracer without tamoxifen administration and the FLEX vectors administered to assess traces of DNA recombination during AAV production, (**A'**) fluorescence images of stained tissue post CBh-FLEX[GFP] and CBh-FLEX[Ascl1-GFP] injection, with white arrows indicating cells where vector or lineage tracer recombination occurs without tamoxifen in MG cells that are not Otx2 or HuC/D positive.

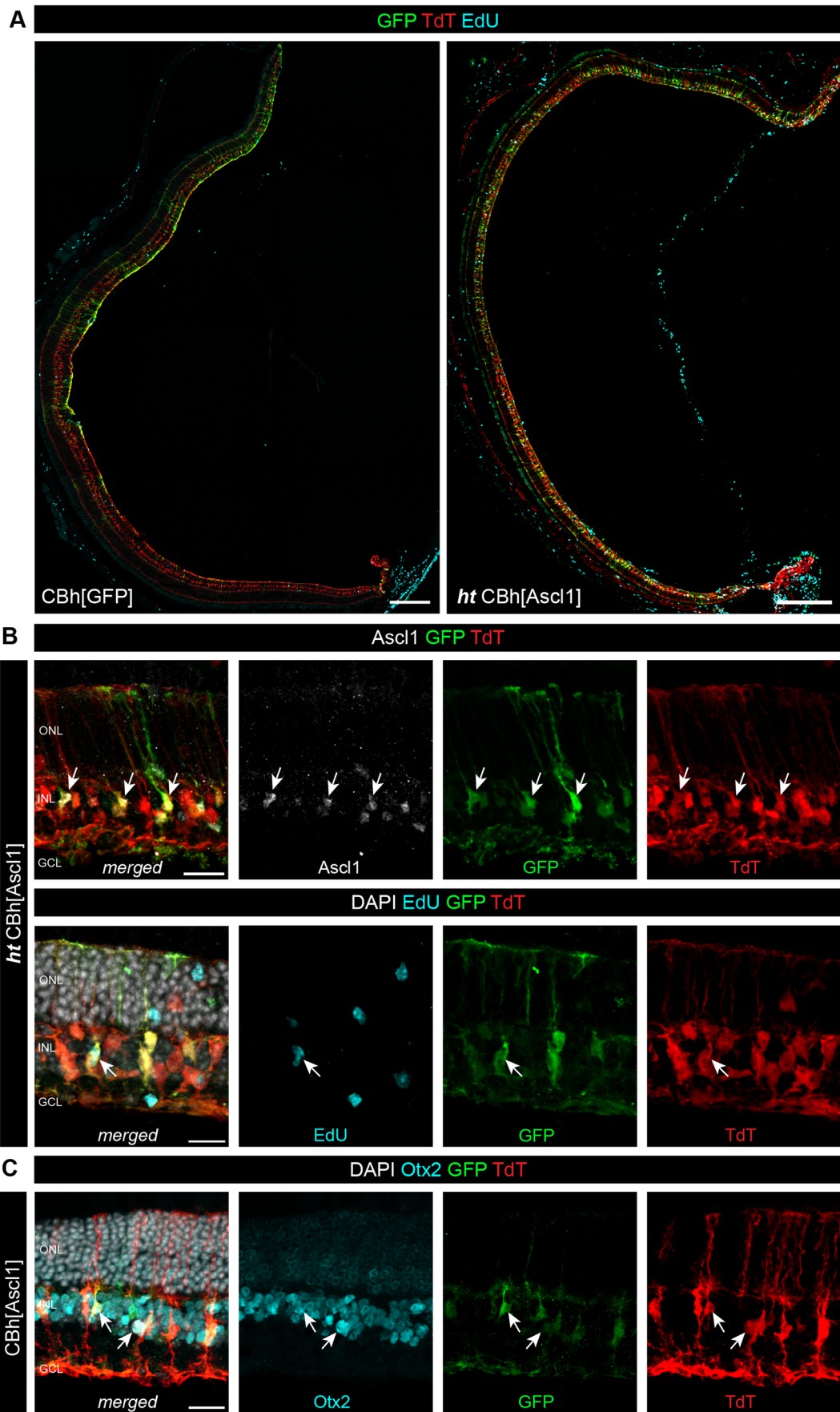

◄ **Figure EV3. AAV transduction and validation of protein markers.**

(**A**) Fluorescence images of Rlbp1-CreERt2 x LSL-TdT central retina cross-sections showing AAV-transduced cells in green, TdT in red and EdU proliferating cells in cyan for control and reprogramming vector conditions, (**B**) fluorescence images after reprogramming with ht AAV/CBh-FLEX[Ascl1], top row: transduced cells (white arrowhead) co-labeled with Ascl1 in white, GFP in green and TdT in red; bottom panel: transduced cells (white arrowhead) co-labeled with EdU in cyan, GFP in green and TdT in red, (**C**) fluorescence images after reprogramming with AAV/CBh-FLEX[Ascl1] showing transduced cells (white arrowhead) co-labeled with Otx2 in cyan, GFP in green and TdT in red; scale bar for (**A**): 200 μm, (**B**, **C**): 20 μm.

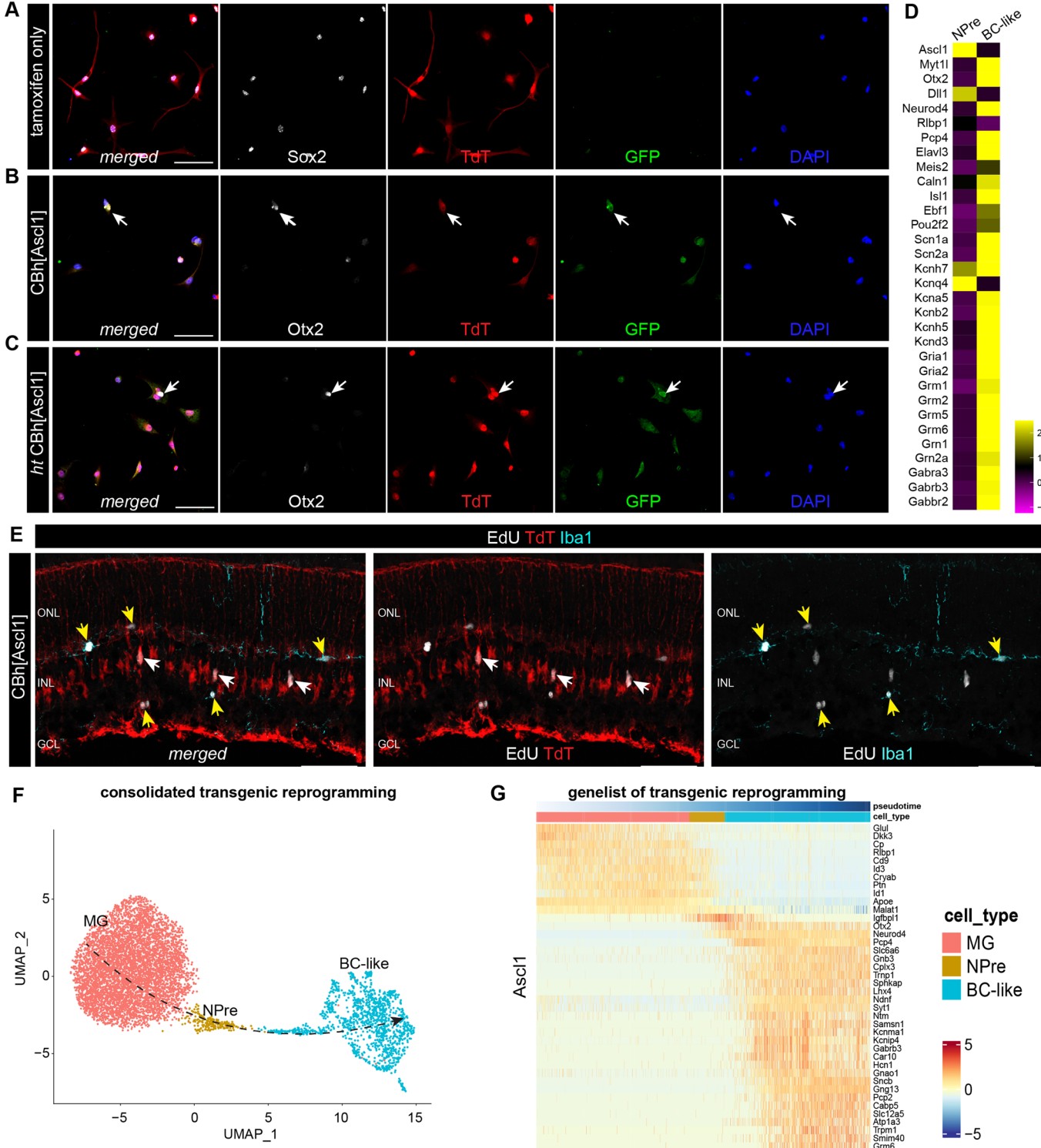

◀ **Figure EV4. Transcriptional analysis of AAV-mediated reprogramming.**

(**A–C**) Fluorescence images of sorted lineage-traced cells on coverslips 24 h post FACS split by condition (white arrows indicate reprogrammed cells) showing merged and single channels of DAPI in blue, GFP vector reporter in green, TdT lineage tracer in red and glial marker Sox2 or bipolar marker Otx2 in white, (**D**) heatmap of average gene expression for selected neuronal genes in clusters NPre and MG-derived neurons following AAV-mediated reprogramming, (**E**) fluorescence images of retinal cross-section with proliferating lineage-traced MG (white arrows) and microglia (yellow arrows) with TdT in red, Iba1 in cyan and EdU in white, (**F**) UMAP of consolidated scRNA-seq data of Ascl1-mediated neurogenesis from transgenic animals (Glast-CreERt2 x LSL-tTA x tetO-Ascl1-GFP), (**G**) heatmap of top 40 differentially expressed genes across pseudotime trajectory from glial cell fate to neuronal cell fate based on scRNA-seq data from (**F**); scale bar: 50 μm, ONL outer nuclear layer, INL inner nuclear layer, GCL ganglion cell layer.

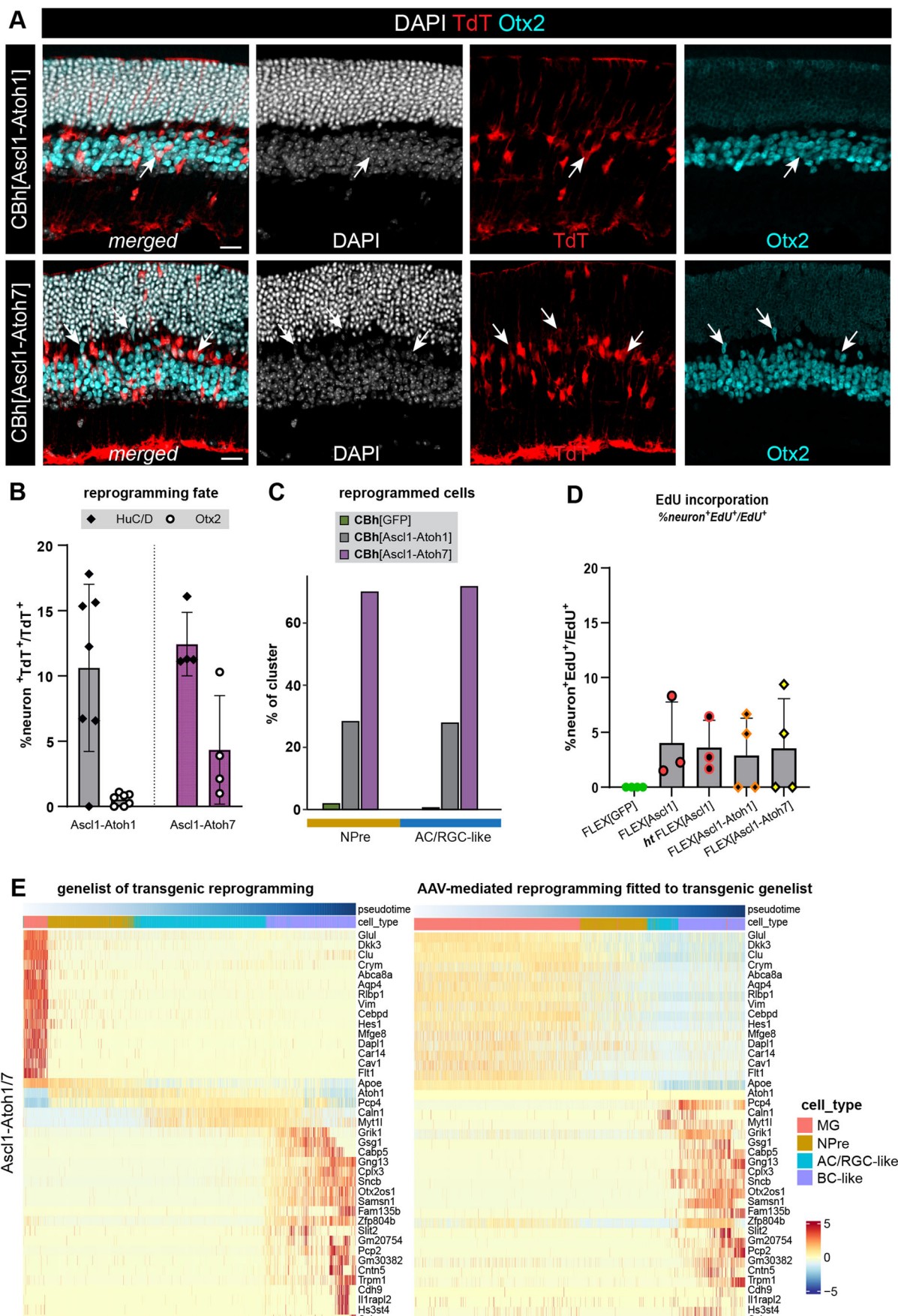

◄

**Figure EV5.   AAV-borne Ascl1-Atoh1/7 expression induces neurogenesis that phenocopies transgenics.**

(**A, A'**) Fluorescence images of reprogrammed cells (white arrows) per condition showing merged and single channels of DAPI in white, TdT in red and Otx2 in cyan, (**B**) bar plot of MG reprogramming to distinct neuronal fates (HuC/D or Otx2) after AAV-mediated Ascl1-Atoh1 or Ascl1-Atoh7 expression counted as a percentage ratio of neuron$^+$TdT$^+$ over all TdT$^+$ cells with each dot being a biological replicate, (**C**) bar plot of the percentrage that each vector treatment contributed to the formation of reprogrammed clusters NPre and MG-derived AC/RGC-like neurons where each column represents a color-coded sample. (**D**) Bar plot of Edu incorporation counted as a ratio of neuron$^+$EdU$^+$ over all EdU$^+$ cells with each dot being a biological replicate, (**E**) heatmap of top 40 differentially expressed genes across pseudotime trajectory from glial cell fate to neuronal cell fate based on scRNA-seq data from transgenic experiments, followed by heatmap of gene expression in scRNA-seq data from AAV experiments that follows the genelist generated from consolidated transgenic data for Ascl1-Atoh1/7 reprogramming; error bars for bar plots:mean plus standard deviation, statistical significance based on ordinary one-way ANOVA with Tukey's multiple comparisons test (significance $P < 0.05$, * = 0.05, ** = 0.01, *** = 0.001, **** = 0.0001, no significance detected); scale bar for (**A–A'**): 20 μm.

                                                           