## [Peer Review File · EMBO Molecular Medicine]

AAV-mediated expression of proneural factors stimulates neurogenesis from adult Müller glia in vivo.

Marina Pavlou, Marlene Probst, Lew Kaplan, Elizaveta Filippova, Aric Prieve, Fred Rieke, and Thomas Reh

Corresponding author: Thomas Reh (tomreh@uw.edu)

Review Timeline:

Submission Date:	13th Sep 24
Editorial Decision:	5th Nov 24
Revision Received:	7th Jan 25
Editorial Decision:	10th Feb 25
Revision Received:	20th Feb 25
Accepted:	21st Feb 25

Editor: Jingyi Hou

Transaction Report:

5th Nov 2024

Dear Tom,

Thank you again for submitting your work to EMBO Molecular Medicine. I apologize for the delay in the review process, which was due to the late arrival of referees' reports. Unfortunately, after a series of reminders, we did not obtain a report from Referee #3. In the interest of time, I have decided to proceed with a decision based on the available reports, rather than prolonging the process further.

As you will see in the reports below, the other two referees find your study of potential interest but have raised several concerns that will need to be thoroughly addressed in a major revision of the manuscript. Since the referees' recommendations are clear, I won't reiterate their points here. It's essential to carefully address all the issues raised by the referees to improve the depth of the analysis and ascertain the authenticity of the reported conversion.

We would welcome the submission of a revised version within three months for further consideration. As you may already know, our editorial policy allows in principle a single round of major revision, and it is therefore essential to provide responses to the referees' comments that are as complete as possible.

Please also contact us as soon as possible if similar work is published elsewhere. If other work is published, we may not be able to extend the revision period beyond three months.

I look forward to receiving your revised manuscript.

Kind regards,
Jingyi

Jingyi Hou
Editor
EMBO Molecular Medicine

We require:

- 1) A .docx formatted version of the manuscript text (including legends for main figures, EV figures and tables). Please make sure that the changes are highlighted to be clearly visible.
- 2) Individual production quality figure files as .eps, .tif, .jpg (one file per figure). For guidance, download the 'Figure Guide PDF': (<https://www.embopress.org/page/journal/17574684/authorguide#figureformat>).
- 3) A .docx formatted letter INCLUDING the reviewers' reports and your detailed point-by-point responses to their comments. As part of the EMBO Press transparent editorial process, the point-by-point response is part of the Review Process File (RPF), which will be published alongside your paper.
- 4) A complete author checklist, which you can download from our author guidelines (<https://www.embopress.org/page/journal/17574684/authorguide#submissionofrevisions>). Please insert information in the checklist that is also reflected in the manuscript. The completed author checklist will also be part of the RPF.

6) It is mandatory to include a 'Data Availability' section after the Materials and Methods. Before submitting your revision, primary datasets produced in this study need to be deposited in an appropriate public database, and the accession numbers and database listed under 'Data Availability'. Please remember to provide a reviewer password if the datasets are not yet public (see <https://www.embopress.org/page/journal/17574684/authorguide#dataavailability>).

12) Author contributions: You will be asked to provide CRediT (Contributor Role Taxonomy) terms in the submission system. These replace a narrative author contribution section in the manuscript.

13) A Conflict of Interest statement should be provided in the main text.

14) Every published paper now includes a 'Synopsis' to further enhance discoverability. Synopses are displayed on the journal webpage and are freely accessible to all readers. They include a short stand first (maximum of 300 characters, including space) as well as 2-5 one-sentences bullet points that summarizes the paper. Please write the bullet points to summarize the key NEW findings. They should be designed to be complementary to the abstract - i.e. not repeat the same text. We encourage inclusion of key acronyms and quantitative information (maximum of 30 words / bullet point). Please use the passive voice. Please attach these in a separate file or send them by email, we will incorporate them accordingly.

15) Include a Reagents and Tools Table as part of the Methods section, which can be downloaded from our author guidelines (<https://www.embopress.org/page/journal/17574684/authorguide#structuredmethods>)

***** Reviewer's comments *****

Referee #1 (Comments on Novelty/Model System for Author):

The overall technical quality of the study is high but the analysis leaves space for improvement.

Re novelty, this is the first study that convincingly shows that AAV can reprogram glia. All previous work before was artefact.

AAVs of course are being explored for gene therapy in vivo.

We are not experts on retinal degeneration but the authors have been using this model for a long time.

Referee #1 (Remarks for Author):

This is a very important study as it demonstrates for the first time in our view that AAVs can indeed reprogram glial cells, in this case Muller glia, into neurons. Previous work making similar claims were subsequently recognised as artefacts. However as the study stands, authors of the previous AAV studies might take this one as a confirmation of theirs. Obviously, it is not the task of the present study to fully demonstrate wherein it differs from the previous work, but it would be helpful to improve the depth of analysis to provide added confidence.

1. While the AAVs encode for GFP, the authors do not show any GFP histology throughout the manuscript. In Figure 1F, is the staining truly Ascl1 or GFP fluorescence? The staining does not look particularly nuclear as one might expect for Ascl1. However, more importantly, by not showing the GFP label of the AAV, one cannot appreciate whether any other cells than MGs express the AAV. This point is of particular importance given the previous widespread use of AAV leading to artefactual labeling of endogenous (but never fate mapped) neurons. Here, the authors succeed in lineage tracing, however, showing us lineage tracing of directly visualised AAV transduced cells would significantly enhance our confidence.

2. The authors make the very important point that Ascl1 (w/o Atohs) promote proliferation. In fact, they show EdU induced neurons which considerably strengthens the notion that these neurons are newly generated. However, it is not clear whether proliferation is a prerequisite. It might difficult to prevent proliferation in vivo to demonstrate its importance for neurogenesis. But it would be useful to show quantification of the relative numbers of reprogrammed cells that incorporated EdU. Given the extensive exposure to EdU it is somewhat surprising that not most of the induced neurons are found to be EdU+. Obviously, proliferation may not be a prerequisite, but it would be helpful to quantify and discuss this at least.

3. Regarding the assignment of identity, it would be helpful to compare the transcriptomes to those of their endogenous counterparts, i.e., bipolar cells, amacrine cells and retinal ganglion cells. Ideally from cells isolated in parallel, but if this is not possible by comparing with pre-existing data sets. Currently, it seems that identity is assigned based only on a few markers. Given maintenance of Sox4 and 11, it seems that induced neurons remain immature (as also suggested by the not very extensive electrophysiology). In any case, we would suggest to rename the induced neurons as X-like rather than X cells.

Minor points:

1. Which cells in the UMAP plots are control cells?

2. Given the importance of the claim, it would be useful to show the expression of some more markers in the newly generated neurons that highlight potential intermediate states along the reprogramming axis.

Referee #2 (Comments on Novelty/Model System for Author):

The authors show no statistical tests e.g. in Figure 1B-D, 3B-C, S3B, so most of their conclusions are actually not supported by statistics. Likewise in Figure 2G they show that 4 of 16 cells respond to light, but they refer to light-responsiveness of the reprogrammed neurons.

The novelty is medium, because the authors themselves have demonstrated reprogramming of Müller Glia to specific types of retina neurons before, but now they are using AAVs to do so. However, for medical relevance it is indeed important, because the authors and others in the field previously used transgenic mice to drive the reprogramming factors, i.e. far off any therapeutic application.

Referee #2 (Remarks for Author):

In this manuscript the authors aim to further develop their previous findings of Müller Glia reprogramming into specific types of retinal neurons by driving the neurogenic reprogramming transcription factor *Ascl1* via AAVs. While this is laudable given the therapeutic aim, the therapeutic relevance is still rather limited, because the viral vector construct is activated by Cre expressed in a transgenic mouse line. As mentioned by the authors, the AAV-driven neurogenic factor expression has been very amenable for artefactual expression in endogenous neurons. The authors aim to avoid this pitfall by using transgenic mice to direct the AAV expression, but by this they sacrifice a big part of the therapeutic applicability. As another label for neurons induced by reprogramming, the authors apply EdU during the reprogramming period to detect dividing cells. Indeed, this shows that only after *Ascl1* expression EdU labelled neurons appear, but they are very few and no statistics are shown to demonstrate significant differences. Likewise, the authors show 4 cells responding to light, but how do we know these are the reprogrammed ones and not some with artefactual reporter activation? The last part compares gene expression data from the previous work of the authors using an entirely transgenic approach to the present AAV and transgenic mouse line approach, showing a considerable overlap. These are interesting data, even though it is irritating that there is no transition between the endogenous Müller glia and the reprogramming ones is visible. Taken together, this manuscript requires considerable improvements to convince the reader as detailed below.

Major suggestions

- 1) Show adequate statistical tests for the data in Figs. 1B-D, 2G (including a control), 3B-C, S3B
- 2) How many of the HuCD+ neurons in Figure 3C are EdU+? Please plot in the same graph and show statistical analysis of these in control and reprogramming condition.
- 3) In Figure 3G and 4A, there is no transition between the endogenous Müller Glia and the reprogramming cells - they are in an entirely different cluster. Maybe earlier analysis could help convincingly show such a transition?
- 4) In Figure 2G the authors show 4 out of 16 reprogrammed cells responding to light. However, they don't show any control and we have no way of knowing if these 4 come from endogenous neurons (by artefactual reporter gene activation) or are indeed reprogrammed. Filling the cells after recording and staining for EdU could help addressing this.

Referee #1 (Comments on Novelty/Model System for Author):

The overall technical quality of the study is high but the analysis leaves space for improvement. Re novelty, this is the first study that convincingly shows that AAV can reprogram glia. All previous work before was artefact. AAVs of course are being explored for gene therapy in vivo. We are not experts on retinal degeneration but the authors have been using this model for a long time.

Referee #1 (Remarks for Author):

This is a very important study as it demonstrates for the first time in our view that AAVs can indeed reprogram glial cells, in this case Muller glia, into neurons. Previous work making similar claims were subsequently recognised as artefacts. However as the study stands, authors of the previous AAV studies might take this one as a confirmation of theirs. Obviously, it is not the task of the present study to fully demonstrate wherein it differs from the previous work, but it would be helpful to improve the depth of analysis to provide added confidence.

We thank the reviewer for acknowledging the importance and rigor of our study. To avoid false confidence in previous artefactual data of other AAV-based studies, we have conducted an additional control experiment in our system that is now summarized in Fig EV2. We asked whether the AAV/FLEX constructs produced any GFP expression in the absence of Cre activity i.e. no tamoxifen administered to transgenic animals. During transgene cloning and vector production, there is a possibility of DNA recombination, which should account for only a fraction of the particles used. We injected FLEX[GFP] and FLEX[Ascl1-GFP] vectors intravitreally into RLBP1-CreERT2 x LNL-TdTomato eyes (n=5 animals) and sacrificed the animals 5 weeks later, consistent with the duration of reprogramming experiments. For the control FLEX[GFP] vector we detected rare GFP+ cells that did not co-localize with neuronal markers Otx2 or HuC/D and had a MG morphology. For the FLEX[Ascl1-GFP] vector we did not detect GFP+ cells (Fig EV2A-A'). In both cases we detected rare Cre-independent TdT+ cells in the red channel that had a MG morphology. This is in line with the sparse leakiness of the RLBP1-CreERT2 x LNL-TdTomato mouse line that we reported when characterizing the lineage tracer (Fig EV1), where a small number of TdT+ cells are detected in the absence of tamoxifen; these TdT+ cells are Sox2+ and do not colocalize with neuronal markers Otx2 or HuC/D (Fig EV1E-E'). Importantly, there were no neurons labeled with either virus in any of the 5 animals we injected, with at least three fields/animal analyzed. Conversely, all transduced GFP+ cells were lineage-traced TdTomato+ MG (representative tile-scan images of central retina section shown in Fig EV3A).

1. While the AAVs encode for GFP, the authors do not show any GFP histology throughout the manuscript. In Figure 1F, is the staining truly Ascl1 or GFP fluorescence? The staining does not look particularly nuclear as one might expect for Ascl1. However, more importantly, by not showing the GFP label of the AAV, one cannot appreciate whether any other cells than MGs express the AAV. This point is of particular importance given the previous widespread use of AAV leading to artefactual labeling of endogenous (but never fate mapped) neurons. Here, the authors succeed in lineage tracing, however, showing us lineage tracing of directly visualised AAV transduced cells would significantly enhance our confidence.

We appreciate the reviewer's assessment and agree that because the Ascl1 antibody used to generate Fig 1F was labelled using the green channel, there is likely some GFP background that's detected as well. As such, we have renamed the label in Fig 1F to "Ascl1-GFP" to acknowledge this phenomenon and added representative histology to confirm Ascl1 protein expression and vector transduction in Fig EV3. In addition to our quantification of transduction efficiency (Fig 1B), we now also show representative tile-scan images of transduced retinas (Fig EV3A), as well as Ascl1+GFP+TdT+ cells and EdU+GFP+TdT+ cells transduced with the FLEX[Ascl1-GFP] vector (Fig EV3B, white arrows). Importantly, we also show histology of transduced and reprogrammed lineage-traced cells that are Otx2+GFP+TdT+ after FLEX[Ascl1-GFP] treatment (Fig EV3C).

2. The authors make the very important point that Ascl1 (w/o Atohs) promote proliferation. In fact, they show EdU induced neurons which considerably strengthens the notion that these neurons are newly generated. However, it is not clear whether proliferation is a prerequisite. It might be difficult to prevent proliferation in vivo to demonstrate its importance for neurogenesis. But it would be useful to show quantification of the relative numbers of reprogrammed cells that incorporated EdU. Given the extensive exposure to EdU it is somewhat surprising that not most of the induced neurons are found to be EdU+. Obviously, proliferation may not be a prerequisite, but it would be helpful to quantify and discuss this at least.

Indeed, we do not know whether proliferation is necessary for neurogenesis. In our previous studies of transgenic-based reprogramming we identified that a subset of the newborn neurons were derived from proliferating MG (~60% of EdU+ cells were HuC/D+ in the Glaxt-CreERT2 x LNL-tTA x tetO-Ascl1-Atoh1 mice, see Fig 3C of (Todd et al., 2021)). While this leaves ~40% that are not EdU+, and suggests that proliferation is not required, it may be that we missed the window for EdU labeling for these cells in previous studies due to the limited temporal availability of EdU following intraperitoneal administrations. In the experiments presented in this study, we find a lower level of EdU+ neurons. We have quantified how many of the EdU+ cells are neurons, as both reviewers requested, and across conditions, we find a mean of 3-4% of EdU+ cells that co-localize with neuronal markers Otx2 or HuC/D (shown in new Fig EV5D). This lower level of EdU+ neurons in the AAV mediated reprogramming may be due to the fact that we are using a different promoter to drive expression in MG (RLBP1 instead of GLAST), and the expression level of reprogramming factors may be different than in the transgenic mice since vector DNA is processed differently than endogenous DNA. Nevertheless, at least some of the MG-derived neurons are labeled with EdU and as such, together with the consistent lineage tracing, we can be confident that we obtain bone fide neurogenesis.

3. Regarding the assignment of identity, it would be helpful to compare the transcriptomes to those of their endogenous counterparts, i.e., bipolar cells, amacrine cells and retinal ganglion cells. Ideally from cells isolated in parallel, but if this is not possible by comparing with pre-existing data sets. Currently, it seems that identity is assigned based only on a few markers. Given maintenance of Sox4 and 11, it seems that induced neurons remain immature (as also suggested by the not very extensive electrophysiology). In any case, we would suggest to rename the induced neurons as X-like rather than X cells.

This was a great suggestion, and we expanded our analysis accordingly in the new Fig 5. We find that the MG-derived neurons from AAV reprogramming show a high degree of overlap with endogenous retinal neurons, particularly of the bipolar cell fate. We also find good overlap in the MG-derived AC/ RGC-like cells with postnatal day 14 mouse retina, suggesting that these classes of MG-derived neurons do not mature as well as the bipolar neurons within the period of time we analyzed. And lastly, we compared the neurons derived from AAV-reprogrammed MG with fetal mouse retina, and find that many of the cells more closely resemble progenitors or neurogenic precursors. These new data are described in the additional text added to the Results:

“To better characterize how similar the reprogrammed cells are to endogenous neurons, we compared the transcriptome of MG-derived neurons from all AAV-infected timepoints with three published retinal datasets (Clark *et al*, 2019; Hoang *et al*, 2020; Li *et al*, 2024). Note, our sorting of TdT+ cells before scRNA-seq excluded endogenous bipolar, amacrine and retinal ganglion cells in our experiments. We compared bipolar subtypes from the Hoang et al dataset from adult mice post NMDA injury and find that the MG-derived BC-like cells upregulate genes that correspond to all BC subtypes in the mouse retina (Fig 5A). We also compared the gene expression of the AC/RGC-like cells with that of mature retinal neurons and find that these are less comparable than the BC-like cells (Fig 5B). However, when we make the same comparisons with P14 retinal neurons, we find neurons derived from the reprogrammed MG overlap well with the bipolar, amacrine and RGC clusters, but not MG (Fig 5C-D). Interestingly, in both the *Ascl1* and *Ascl1-Atoh1/7* reprogrammed MG, there is a cluster of cells that do not align with any of the retinal neurons at P14. When these cells are instead aligned with the E14 retina, they align primarily with the progenitors, and transitioning cells (T0 and T1) (Fig 5E-F). This solidifies our conclusion that vector-mediated expression of proneural genes in adult mouse MG can stimulate neurogenesis of distinct neuron classes, though not all these MG-derived neurons fully differentiate within the period of time we have analyzed.”

Minor points:

1. Which cells in the UMAP plots are control cells?

If the reviewer is referring to the UMAP in Fig 2B, this is an integrated UMAP of sorted and sequenced TdT+ cells from all four vector conditions: CBh-FLEX[GFP], CBh-FLEX[*Ascl1*-GFP], Ef1 α -FLEX[*Ascl1*-GFP], *ht* CBh-FLEX[*Ascl1*-GFP]. Sorted cells from animals treated with the CBh-FLEX[GFP] vector were considered control cells and we plotted the distribution of MG, reactive MG and reprogrammed cells across treatments in this integrated object to demonstrate neurogenesis only occurred with *Ascl1* treatment. If the reviewer considers control cells to be endogenous neurons, please see our new Fig 5 and previous response above.

2. Given the importance of the claim, it would be useful to show the expression of some more markers in the newly generated neurons that highlight potential intermediate states along the reprogramming axis.

The matter of intermediate states is indeed interesting. Our staining with marker *Pcp4* intended

to address this point (Fig 4K) in support of our observations that AAV-mediated reprogramming, with our protocol, has sharp transitions from NPre to neurons. From our scRNAseq data, we detected transcripts of neuronal markers expressed early during neurogenesis, including Pcp4 (Fig EV4D). The pattern of Pcp4 expression was interesting to us because it differed between transgenic reprogramming and AAV-mediated reprogramming. In the former, Pcp4 is first downregulated as the cells are in a glial state, then upregulated as they become progenitors and again downregulated as they become BC-like (Fig EV5E). This is not the case with AAV reprogramming, as Pcp4 seems to come on sharply when cells become BC-like (Fig 2E, Fig EV4G and Fig EV5E). Capturing both EdU+Otx2+ and EdU+Pcp4+ cells (Fig 4K) confirms that when we reprogram with AAV/Ascl1 or AAV/Ascl1-Atoh1/7 the new neurons are unlikely to remain in an intermediate state.

Referee #2 (Comments on Novelty/Model System for Author):

The authors show no statistical tests e.g. in Figure 1B-D, 3B-C, S3B, so most of their conclusions are actually not supported by statistics. Likewise in Figure 2G they show that 4 of 16 cells respond to light, but they refer to light-responsiveness of the reprogrammed neurons. The novelty is medium, because the authors themselves have demonstrated reprogramming of Müller Glia to specific types of retina neurons before, but now they are using AAVs to do so. However, for medical relevance it is indeed important, because the authors and others in the field previously used transgenic mice to drive the reprogramming factors, i.e. far off any therapeutic application.

We appreciate the reviewer's evaluation and agree that our study is a step towards translation but not a ready-to-use regenerative therapy yet. Given that several studies have reported artefactual data of vector-mediated reprogramming using viral vectors alone, we sought to dissect the question of vector-based neurogenesis and demonstrate concretely that neurogenesis is possible with robust lineage-tracing of transduced MG. We consider our work an important contribution to the field, as our study shows for the first time that neurogenesis is possible when proneural factors are expressed from an AAV vector *in vivo*.

Referee #2 (Remarks for Author):

*In this manuscript the authors aim to further develop their previous findings of Müller Glia reprogramming into specific types of retinal neurons by driving the neurogenic reprogramming transcription factor *Ascl1* via AAVs. While this is laudable given the therapeutic aim, the therapeutic relevance is still rather limited, because the viral vector construct is activated by *Cre* expressed in a transgenic mouse line. As mentioned by the authors, the AAV-driven neurogenic factor expression has been very amenable for artefactual expression in endogenous neurons. The authors aim to avoid this pitfall by using transgenic mice to direct the AAV expression, but by this they sacrifice a big part of the therapeutic applicability. As another label for neurons induced by reprogramming, the authors apply EdU during the reprogramming period to detect dividing cells. Indeed, this shows that only after *Ascl1* expression EdU labelled neurons appear, but they are very few and no statistics are shown to demonstrate significant differences. Likewise, the authors show 4 cells responding to light, but how do we know these are the reprogrammed ones and not some with artefactual reporter activation? The last part compares gene expression data from the previous work of the authors using an entirely transgenic approach to the present AAV and transgenic mouse line approach, showing a considerable overlap. These are interesting data, even though it is irritating that there is no transition between the endogenous Müller glia and the reprogramming ones is visible. Taken together, this manuscript requires considerable improvements to convince the reader as detailed below.*

The reviewer raises several good points which we address in our revised version. Please see our detailed responses.

Major suggestions

1) Show adequate statistical tests for the data in Figs. 1B-D, 2G (including a control), 3B-C, S3B

We implemented the author's suggestion and have now included statistical analyses of our data, after performing one-way ANOVA and Tukey's multiple comparisons across conditions. We updated the following bar graphs with only the statistically significant relationships: Figure 1B, 1C, 1D, 3B, 3C. The proliferation and reprogramming effects between our control vector FLEX[GFP] and the low-titer FLEX[Ascl1] vectors was not statistically significant, although the high-titer FLEX[Ascl1] had a significant proliferation and reprogramming effect. This is likely due to the significantly lower transduction of the low-titer FLEX[Ascl1] vectors compared to the control FLEX[GFP] and the high-titer FLEX[Ascl1] (Fig 1B). We also show significant reprogramming with FLEX[Ascl1-Atoh1/7] vectors compared to the control FLEX[GFP] (Fig 3C), with FLEX[Ascl1-Atoh7] having a significant proliferation effect compared to both the control and FLEX[Ascl1-Atoh1] vectors (Fig 3B). We have also expanded our ephys data in Fig 2G and now include recordings from GFP- endogenous neurons as control. For more details, please see our response to comment 4.

2) How many of the HuCD+ neurons in Figure 3C are EdU+? Please plot in the same graph and show statistical analysis of these in control and reprogramming condition.

We have now expanded our analyses and include a new graph (Fig EV5D) where we calculated how many of the EdU+ cells co-localize with a neuronal marker (neuron+) Otx2 or HuC/D. Only a small number of new neurons are EdU+ across treatments (on average 3-4%) and in some cases none at all, which suggests that cell-cycle re-entry is not required for reprogramming. Only a subset of reprogrammed neurons originates from proliferating progenitors, which is a phenomenon we report also in our previously published transgenic experiments. As reviewer 1 states, it is unclear whether MG proliferation is necessary for neurogenesis in the adult mammalian retina, which is why we rely on transgenic lineage-tracing to record reprogramming and quantify MG proliferation as an independent outcome measure. We interpret the sparser EdU incorporation in newborn neurons as indirect reprogramming of MG to proliferating progenitors and then into neurons, which is not the path for every reprogramming event both in our transgenic experiments and AAV-experiments.

3) In Figure 3G and 4A, there is no transition between the endogenous Müller Glia and the reprogramming cells - they are in an entirely different cluster. Maybe earlier analysis could help convincingly show such a transition?

The reviewer makes the point that in prior publications in which we use transgenic animals to reprogram MG into neurons, the UMAP shows a continuum between clusters as cells transition, from MG to neurogenic precursors to neurons (such as in Fig EV4F), and wonders why this is not present in the AAV-reprogrammed MG experiments. The difference could be an effect of cell number; in transgenic animals even though between 20 and 80 of the MG became neurogenic, depending on the TF combination, the number of cells in these transition populations was only a fraction of the reprogrammed cells; by contrast, when we reprogram the MG using AAVs we obtain 1-10% reprogramming. As a result, we may not pick up sufficient numbers of transition cells to show a connecting cluster in the UMAP space. Even so, there are some MG cells

proximal to the NPre cluster (Fig 2B and circled in figure below) as well as a faint branching point of reprogrammed clusters from the MG cluster in Fig 4B where we combined the scRNAseq data from all timepoints. Although transition states are not captured perfectly in our initial analysis, we have now done additional comparisons of the transcriptomes post AAV-reprogramming and the developing E14 and P14 mouse retina (Figure 5). There, we do see a clear overlap of reprogrammed cells with progenitors (Prog), transitioning cells (T0 & T1) and the clusters of early photoreceptors and amacrine cells in the developing E14 retina (Fig 5C-C').

4) In Figure 2G the authors show 4 out of 16 reprogrammed cells responding to light. However, they don't show any control and we have no way of knowing if these 4 come from endogenous neurons (by artefactual reporter gene activation) or are indeed reprogrammed. Filling the cells after recording and staining for EdU could help addressing this.

We thank the reviewer for this suggestion and have now included additional control data. To address concerns over artefactual AAV expression in endogenous neurons, we now added further control data summarized in new EV Figures 2 and 3. We demonstrate that the vector cargo is not expressed in endogenous Otx2+ or HuC/D+ neurons in the absence of tamoxifen (Fig EV2) and that transduced GFP+ cells co-localize with lineage-traced TdTomato+ MG (representative tile-scan images of central retina section shown in Fig EV3A). As such, the only GFP+ neurons we detect and could record from during patch-clamp experiments are from lineage-traced MG (post tamoxifen).

We have also added new ephys data from endogenous neurons (GFP- open circle datapoints) to the summary plot in Figure 2G. Light responses of endogenous neurons in retinal slices are quite variable (likely from damage introduced in slicing). While reprogramming caused some cells to generate light responses similar to the most robust responses measured in endogenous neurons, none of the MG-derived cells shared overlapping electrical properties with the endogenous neurons. This is clear from the lack of overlap along the y-axis, which measured the linearity of a cell's current-voltage relationship. Activation of voltage-activated channels, especially K+ channels, often cause the resistance to decrease as a cell depolarizes. This causes the current-voltage relation to be nonlinear (examples of 4 endogenous neuron

recordings in figure below). Glia, with a mostly voltage-independent K⁺ conductance, have quite linear current-voltage relations. The comparison of current-voltage relation hence indicates that none of the MG-derived GFP⁺ cells express a complement of channels identical to that of endogenous neurons, despite some of those cells having large light responses. Staining for EdU would indeed be a nice addition to this analysis. Unfortunately, this would be extremely challenging since most of the cells do not survive removal of the patch electrode at the end of the recording. It is worth noting that a substantial fraction of the GFP⁺TdT⁺ cells had light responses (8 of 19, 4 of which were small like the α example in Figure 2G).

GFP⁺ cells: voltage response to current steps

10th Feb 2025

Dear Prof. Reh,

Thank you for submitting your revised manuscript to EMBO Molecular Medicine. We have now received the enclosed report from Reviewer #1 who agreed to re-assess your work. Since the original Reviewer #2 is unable to re-review the manuscript, and given that the feedback from Reviewers #1 and #2 during the the previous round were aligned, we asked Reviewer #1 to also review the your responses to Reviewer #2's main points.

As you will see, Reviewer #1 thinks that all concerns of the two reviewers have been adequately addressed. Therefore, I am pleased to inform you that we will be able to accept your manuscript pending the following amendments:

1. Please remove the Authors' Contribution section from the manuscript file.
2. Please provide up to five keywords.
3. Author checklist: please enter corresponding author name, journal name and manuscript ID.
4. Please note that the funding information needs to be part of Acknowledgements section.
5. "Materials and Methods" should be renamed to " Methods".
6. "competing interests" should be renamed to "Disclosure Statement & Competing Interests".
7. "Data and materials availability" should be renamed to " Data availability".
8. Please remove "short description" from the manuscript file.
9. The manuscript sections should be in the following order: Title page - Abstract & Keywords - Introduction - Results - Discussion - Methods - Data Availability - Acknowledgments - Disclosure Statement & Competing Interests - References - Figure Legends -Main Tables with legends - Expanded View Figure Legends.
10. Include a Reagents and Tools Table, which can be downloaded from our author guidelines (<https://www.embopress.org/page/journal/17574684/authorguide#structuredmethods>)
When submitting your revised manuscript, please do not include the Reagents and Tools Table in the Methods section of the manuscript but upload it as a separate file choosing the file type "Reagent Table". Please remove the current Table 1 and Table 2.
11. The paper explained: EMBO Molecular Medicine articles are accompanied by a summary of the articles to emphasize the major findings in the paper and their medical implications for the non-specialist reader. Please provide a draft summary of your article highlighting
 - the medical issue you are addressing,
 - the results obtained and
 - their clinical impact.This may be edited to ensure that readers understand the significance and context of the research. Please refer to any of our published articles for an example.
12. Every published paper now includes a 'Synopsis' to further enhance discoverability. Synopses are displayed on the journal webpage and are freely accessible to all readers. They include a short stand first (maximum of 300 characters, including space) as well as 2-5 one-sentences bullet points that summarizes the paper. Please write the bullet points to summarize the key NEW findings. They should be designed to be complementary to the abstract - i.e. not repeat the same text. We encourage inclusion of key acronyms and quantitative information (maximum of 30 words / bullet point). Please use the passive voice. Please attach these in a separate file or send them by email, we will incorporate them accordingly.

Please also suggest visual abstract to illustrate your article as a PNG file 550 px wide x 300-600 px high.
13. Source data: the source data for Fig1B, Fig 2F, Fig 3B,C is requested by our source data officer but is missing. Please provide it in the next submission.

14. Please address the following comments related to figure legends:

- Please note that for the figures 1B, C, D; 3B, C p-values and statistical tests are indicated in the legends. However, comparison for the same, "****/**/**/" has not been represented in the figures. Please rectify this in the figures or legends as applicable.
- Please note that the box plots need to be defined in terms of minima, maxima, centre, bounds of box and whiskers, and percentile in the legends of figures EV1 B.
- Please note that the measure of center for the error bars needs to be defined in the legends of figures 1B, C, D; 3B, C; EV5 B, D.
- Please note that the white arrows are not defined in the legend of figures 2F, EV2 A', EV4 A-C. This needs to be rectified.
- Please note that the red arrows are not defined in the legend of figures 3E, F. This needs to be rectified.

I look forward to seeing a revised form of your manuscript as soon as possible.

Kind regards,
Jingyi

Jingyi Hou
Senior Editor
EMBO Molecular Medicine

*** Instructions to submit your revised manuscript ***

***** Reviewer's comments *****

Referee #1 (Comments on Novelty/Model System for Author):

Statistical analysis has been revised as requested by referee #2.

Main novelty consists of the evidence for successful Müller glia-to-neuron conversion by AAVs. This is important in the context that much published work using AAVs in the past has been artefactual. Here, artefact has been convincingly ruled out which is an important step for the field. This is also a step towards devising a strategy for employment of Müller glia reprogramming for clinical treatment of eye disease. The model used is a standard model to study retinal repair.

Referee #1 (Remarks for Author):

All concerns of the two referees were adequately addressed by performing new experiments and analyses and where suggested rephrasing of the text to best represent the data.

The authors addressed the remaining editorial issues.

21st Feb 2025

Dear Prof. Reh,

We are pleased to inform you that your manuscript is accepted for publication and is now being sent to our publisher to be included in the next available issue of EMBO Molecular Medicine.

Yours sincerely,
Jingyi

Jingyi Hou
Senior Editor
EMBO Molecular Medicine
